



# Establishing the Impact of Model Surfactants on Cloud Condensation Nuclei Activity of Sea Spray Aerosols

Sara D. Forestieri[1,#], Sean M. Staudt[2], Thomas M. Kuborn[2], Katharine Faber[3], Christopher R. Ruehl[4], Timothy H. Bertram[2], Christopher D. Cappa[1, *]

[1]Dept. of Civil and Environmental Engineering, University of California, Davis, CA, USA

[2]Dept. of Chemistry, University of Wisconsin, Madison, WI, USA

[3]Dept. of Chemistry, Carleton College, Northfield, MN, USA

[4]California Air Resources Board, Sacramento, CA, USA

* Correspondence to: E-mail: cdcappa@ucdavis.edu

[#] *Now at: California Air Resources Board, Sacramento, CA, USA*

**Abstract.** Surface active compounds present in aerosols can increase their cloud condensation nuclei (CCN) activation efficiency by reducing the surface tension ($\sigma$) in the growing droplets. However, the importance of this effect is poorly constrained by measurements. Here we present

estimates of droplet surface tension near the point of activation derived from direct measurement of droplet diameters using a continuous flow stream-wise thermal gradient chamber (CFSTGC). The experiments used sea spray aerosol mimics composed of NaCl coated by varying amounts of (i) oleic acid, palmitic acid or myristic acid, (ii) mixtures of palmitic acid and oleic acid, and (iii) oxidized oleic acid. Significant reductions in $\sigma$ relative to that for pure water were observed for

these mimics at relative humidity (RH) near activation (~99.9%) when the coating was sufficiently thick. The calculated surface pressure ($\pi = \sigma_{H2O} - \sigma_{observed}$) values for a given organic compound or mixture collapse onto one curve when plotted as a function of molecular area for different NaCl seed sizes and measured RH. The observed critical molecular area ($A_0$) for oleic acid determined from droplet growth was similar to that from bulk experiments conducted in a

Langmuir trough. However, the observations presented here suggest that oleic acid in microscopic droplets may exhibit larger $\pi$ values during monolayer compression. For myristic acid, the observed $A_0$ compared well to bulk experiments on a fresh subphase, for which dissolution has an important impact. A significant kinetic limitation to water uptake was observed for NaCl particles coated with pure palmitic acid, likely as a result of palmitic acid



being able to form a solid film.  However, for binary palmitic acid-oleic acid mixtures there was no evidence of a kinetic limitation to water uptake. Oxidation of oleic acid had a minor impact on the magnitude of the surface tension reductions observed, potentially leading to a slight reduction in the effect compared to pure oleic acid. A cloud condensation nuclei (CCN) counter

was also used to assess the impact on critical supersaturations of the substantial σ reductions observed at very high RH. For the fatty acid-coated NaCl particles, when the organic fraction ($\epsilon_{org}$) was >0.90 small depressions in critical supersaturation were observed. However, when $\epsilon_{org}$ < 0.90 the impact on critical supersaturation was negligible. Thus, for the fatty acids considered here, the substantial σ reductions observed at high RH values just below activation have limited

impact on the ultimate critical supersaturation. A surface film model is used to establish the properties that surface active organic molecules must have if they are to ultimately have a substantial impact on the activation efficiency of SSA. To influence activation, the average properties of surface-active marine-derived organic molecules must differ substantially from the long chain fatty acids examined, having either smaller molecular volumes or larger molecular

areas. The model results also indicate that organic-driven surface tension depression can serve to buffer the critical supersaturation against changes to the organic-to-salt ratio in particles in which the organic fraction is sufficiently large.

## 1    Introduction

Surface active organic matter present in atmospheric aerosols has received considerable

attention given its important role in heterogeneous chemistry (Knopf et al., 2005;Shaloski et al., 2017), aerosol water uptake and evaporation (Davies et al., 2013), and potential impact on the ability of particles to activate into cloud droplets (Ruehl et al., 2016;Ovadnevaite et al., 2017). Amphiphilic, surface active species have the potential to lower surface tension of a growing droplet, relative to pure water, due to their presence at the air-water interface. Through the

Kelvin effect, this reduction in surface tension can, in theory, reduce the critical supersaturation, thereby increasing particle cloud condensation nuclei activation efficiency (Farmer et al., 2015).

The importance of surface tension depression, however, remains contentious given the difficulty in quantifying surface tension in microscopic droplets directly (Broekhuizen et al., 2006;Broekhuizen et al., 2004;Mircea et al., 2005;Henning et al., 2005;Wex et al., 2009;Gray Bé

et al., 2017;Good et al., 2010a;Good et al., 2010b;Jurányi et al., 2010;Fuentes et al.,





2011;Duplissy et al., 2008;Prisle et al., 2011;Petters and Kreidenweis, 2013;Sareen et al., 2013;Giordano et al., 2013;Moore et al., 2008;Li et al., 1998). Surface tension reductions have been observed in bulk solutions of organic matter extracted form aerosols when the organic concentration is sufficiently large (Facchini et al., 1999;Noziere et al., 2014). Calculations of

critical supersaturation based on such measurements indicate a strong potential for organic compounds to influence cloud formation and properties (Facchini et al., 1999;Mircea et al., 2002). There have also been direct single particle surface tension measurements for aerosols at subsaturated humidities (Lee et al., 2017). However, it has been suggested that the effects of surface tension lowering on cloud droplet activation are offset by depletion of solutes from the

bulk as they partition to the surface of microscopic droplets having high surface area-to-volume ratios (Prisle et al., 2010;Asa-Awuku et al., 2009;Sorjamaa et al., 2004).

One challenge in assessing the potential influence of surface tension depression on CCN activity has been the lack of unambiguous evidence for the impact of surface tension on droplet activation. Recent work, using a custom-built continuous flow thermal gradient chamber

(CFSTGC) to measure the sizes of droplets around 100% RH, has allowed for more direct estimates of the surface tension depression in microscopic droplets at humidities relevant to cloud droplet formation. The size of droplets around 100% RH is sensitive to deviations in surface tension from that of pure water, much more so than at lower relative humidities. Thus, droplet size measurements can be used to infer surface tension values. Using this method, it was

determined that the surface tension of droplets was substantially reduced below that of pure water at RH ~ 100% for binary mixtures of individual surface active compounds and ammonium sulfate and for NaCl coated with secondary organic aerosol when the coatings were sufficiently thick (Ruehl and Wilson, 2014;Ruehl et al., 2012). The reduction in surface tension at a given RH caused the droplets to grow larger than predicted if the surface tension was assumed to be

that of pure water (Ruehl et al., 2016).

One particle type where surface tension impacts on cloud droplet activation might be of particular importance is with sea spray aerosol (SSA) particles. There is indirect evidence that surface tension reductions affect the efficiency of SSA particle activation. For example, Collins et al. (2016) observed high hygroscopicity parameters ($\kappa > 0.7$) for small SSA particles (<150

nm) generated during a suite of microcosm phytoplankton bloom, up to very high chlorophyll-a





concentrations in the source water (a marker for biological activity), yet no correlation between κ and biological activity was observed. Although Collins et al. (2016) did not measure the particle composition it seems likely that the organic fraction of the particles was large due to the high biological activity (O'Dowd et al., 2004b). Additionally, very large organic fractions were

separately observed for small particles (< 200 nm) for a similar microcosm experiment (Deane et al., Submitted). The potential for substantial surface tension lowering was invoked to explain the high κ values observed. However, Fuentes et al. (2011) performed experiments using seawater enriched in marine exudates and found no evidence that surface tension reduction impacted CCN activation, observing instead a slight decrease in the CCN activation efficiency with increasing

organic content, although the measured organic fractions of their particles were relatively low (<40%) and therefore would not to enhance CCN activity if this organic matter formed a compressed film on the droplet surface (Ruehl et al., 2016).

To better understand the impact of surface active organic species on SSA particle activation efficiency, and on droplet activation in general, we report on measurements of droplet

sizes at RH values just below activation for NaCl particles coated with varying amounts of marine-relevant organic compounds, specifically long-chain fatty acids. These experiments were done as part of the MadFACTS campaign (Madison Fatty Acid Coating Thickness Study). Long-chain fatty acids are an important class of organic compounds found in submicron SSA particles (Cochran et al., 2017;Cochran et al., 2016). Since this class of organic species is surface active

(Schwier et al., 2012) they have the potential to enhance observed CCN activation efficiency by depressing surface tension, but the overall effect of fatty acid addition on CCN activation efficiency of salt particles has been shown to be minimal (Nguyen et al., 2017). In this work we will clarify why such highly surface active species have little impact CCN activation efficiency. Additionally, since particles in the ambient atmosphere are composed of complex mixtures of

organic species, the role of mixtures was investigated by comparing single component and binary surfactant systems and by comparing oxidized and unoxidized systems. These measurements were used to estimate the surface tension of the droplets as a function of the relative abundances of NaCl and the organic component(s) at a given RH, or as a function of RH for a fixed dry particle composition. We connect these near-activation surface tension measurements to the

surface tension estimated at activation with a traditional CCN counter to assess whether the





observed reductions in surface tension significantly affect critical supersaturations for these chemical systems.

## 2 Methods

A suite of instrumentation was used to generate sea spray aerosol mimics and monitor
their hygroscopic properties and chemical composition. A general schematic of the experimental
set-up is shown in Figure 1.

### 2.1 Particle Generation and Processing

Polydisperse NaCl particles were generated from a constant output atomizer (TSI Inc.)
containing a 0.05 M solution of NaCl (99% purity). The air stream was dried with a diffusion
denuder to < 20% RH. The particles were size selected according to their mobility diameters
ranging from 150 nm to 200 nm. The sheath to aerosol flow ratio was 10:1 to provide a relatively
narrow distribution. The monodisperse particles were passed through an oven containing an
aluminum sample holder with surface-active organic compounds. The organic compounds used
were: myristic acid (99% purity; Sigma Aldrich), palmitic acid (99% purity; Sigma Aldrich), and
oleic acid (90% or 99% purity; Sigma Aldrich). Myristic acid and palmitic acid are saturated
fatty acids containing 14 and 16 carbons, respectively, and exist as waxy solids at room
temperatures. Oleic acid is an unsaturated fatty acid that exists as a liquid at room temperature
and contains 18 carbons. For mixed surfactant studies, a sample holder containing known
amounts of each organic compound (molar ratio 1:1) was added to the oven. In the oven, the
organic compounds evaporated into the vapor phase. Upon exiting the oven, the air stream
cooled and the organic compounds condensed onto the NaCl particles resulting in surfactant-
coated NaCl particles. Oven temperatures ranged from 85°C for thin coatings to 130°C for thick
coatings. A charcoal denuder was placed after the oven to remove residual vapor from the air
stream. A second DMA (TSI Inc.; Model 3071) was used to select particles of a given size from
the coated distribution and served to remove small nucleated particles composed purely of the
organic compound(s). Note that for the second DMA the neutralizer (Kr-85) was bypassed,
ensuring that only particles containing NaCl were sampled. Flow was then split isokinetically
through a flow splitter (TSI Inc.; model 370800) to the various instrumentations.

For oxidation experiments, oleic acid coated NaCl particles were exposed to $O_3$ in a
vertical glass flow tube. Particles were introduced into the flow tube via a side port located 19



cm below the top of the tube and exited through a side port 77 cm below the inlet. Ozone was injected through a moveable stainless steel injector that passed through the top of the flow tube. The exit of the injector was positioned between the particle inlet and exit ports, hereafter referred to as the active region. The residence time was varied by moving the injector position and

calculated based on the distance between the injector and the flow tube exit. Total flow through the flow tube was 1.8 SLPM, with 0.5 and 1.3 SLPM for the $O_3$ and particle streams, respectively. The injected $O_3$ concentration was 1 ppm resulting in an estimated concentration of 278 ppb in the flow tube, after dilution. Residence times calculated based on the flow rate and flow tube volume varied from 0 to 57 s. However, the air was sampled from the side of the flow

tube and so the effective residence time was longer and was never as short as 0 s. There was a bypass channel for the flow tube to monitor particle size and hygroscopicity prior to the flow tube experiments. The estimated ozone exposure ranged from $1.4 \times 10^{14}$ to $3.9 \times 10^{14}$ molecules cm$^{-3}$ s for non-zero residence times. These are likely lower limits to the true $O_3$ exposure, given the non-ideal flow within the flow tube resulting from the side sampling. Also note that a

stainless steel injector was used, and thus the O3 concentration in the flow tube was likely somewhat lower than the estimated value.

## 2.2 Hygroscopicity Measurements

Wet droplet diameter distributions at RH values near 100% (both above and below) were measured by a CFSTGC, described in detail by Ruehl et al. (2010). Briefly, particles are

humidified in a temperature-controlled tube lined with wetted filter paper (102 cm in length, with an effective inner diameter of 2.2 cm). The temperature gradient across the length of the tube could either be positive or negative allowing for both sub-and super-saturated RH values to be achieved (Roberts and Nenes, 2005), with a working range from 99.8% to 100.06%. Before exiting the CFSTGC, wet droplet size distributions and velocity were measured with a phase-

doppler interferometer (PDI; Artium Technologies, Inc.). The particular configuration of this CFSTGC instrument allows for accurate and precise determination of droplet diameters at a known RH value prior to activation. The total flow through the chamber was 0.5 SLPM, with a sheath:sample flow ratio of 2.33. The total residence time of the chamber centerline was ~22 seconds. The mode diameter for each measurement was obtained by fitting the number-weighted

wet droplet distribution to a Gaussian curve using a function in the data processing program Igor (Wavemetrics, v6.37), with each scan consisting of ~1,000 droplet measurements. Average



temperatures in the CFSTGC ranged from 20 °C to 25 °C. The absolute temperatures were varied from day to day due to fluctuations in the room temperature, and were selected to prevent condensation in the detection chamber. However, since the instrument RH depends primarily on ΔT and not absolute T, these variations do not affect instrument performance.

RH in the CFSTGC was calibrated by sampling size-selected salt particles composed of either NaCl or ammonium sulfate. The κ-Köhler equation (Petters and Kreidenweis, 2007) was then used to calculate RH

$$\frac{RH}{100} = \frac{D_{wet}^3 - D_{dry}^3}{D_{wet}^3 - D_{dry}^3(1-\kappa_{avg})} \exp\left[\frac{4\sigma V_{H_2O}}{kTD_{wet}}\right] \tag{1}$$

where $D_{wet}$ is the measured wet diameter, $D_{dry}$ is the dry diameter, κ is the known hygroscopicity parameter, σ is surface tension, $V_{H2O}$ is the molar volume of water in the droplet, $k$ is the Boltzmann constant, and $T$ is temperature. The first term corresponds to the Raoult effect, which accounts for reductions in water activity to the dissolution of solutes. The effective solubility of a given species is parameterized by the κ parameter, which ranges here from ~0 (insoluble) to 1.4

(very soluble). The exponential, or Kelvin, term accounts for the enhanced vapor pressure over a curved surface and is proportional to σ. For the calibrations, σ is assumed to be equal to that of water. This is reasonable given the dilute concentrations of salt (~0.05 M) in the aqueous droplets and the lack of surface-active species. The accuracy of RH calibrations depends on the assumed κvalues and shape-correction factors of the calibration salts. The RH values were

calculated from Eqn. 1 assuming that κ values were 1.3 and 0.61 and shape-correction factors were 1.08 and 1.04 for NaCl and ammonium sulfate, respectively. As such, the uncertainty for RH in the CFSTGC was characterized by calibrating the chamber with both size-selected NaCl and ammonium sulfate, switching between the two salts every 3-5 minutes. Results of this comparison are shown in Figure S1. The slope of the linear fit was 1.00, with the majority of the

data falling within 0.025% of the 1:1 line. During experiments with NaCl particles coated with organic compounds, the RH was calibrated multiple times a day with either never-coated particles or with NaCl particles where the organic coating was completely removed by passing the particles through a thermodenuder at 250 °C. (No difference was found between never-coated and thermally denuded NaCl particles.) By calibrating throughout the day noise due to RH drift



was minimized. Most experiments were conducted holding both RH and the NaCl seed size constant while varying the amount of organic coating. For one experiment with oleic acid as a coating, the composition (or coating amount) of the particles was held constant, while RH was varied to characterize how droplet size varied leading up to the point of activation. In this

manner, a Köhler curve is mapped out (Ruehl et al., 2016).

For a subset of experiments, the number of particles that activate into cloud droplets at a given supersaturation was characterized with a CCN counter (CCNC; Droplet Measurement Technologies, Model CCN-100) (Roberts and Nenes, 2005). Total particle concentrations were measured concurrently with a condensation particle counter (CPC; TSI Inc. Model 3787). The

combination of these measurements allowed for the calculation of the fraction of activated particles ($f_{act}$) as a function of supersaturation ($s$ = RH/100 - 1), with scanned supersaturation values ranging from 0.03% to 0.1%. The critical supersaturation ($s_c$) was determined by fitting a sigmoidal function to $f_{act}$ versus $s$. The apparent hygroscopicity parameter (Petters and Kreidenweis, 2007) was then calculated using the following equation:

$$\kappa_{app} = \frac{4A^3}{27\,D_{dry}^3 \ln^2 S_c} \quad \text{and} \quad A = \frac{4\sigma M_W}{RT\rho_W} \tag{2}$$

where $Sc = s_c + 1$, $M_W$ is the molecular weight of water, $D_{dry}$ is the dry diameter, $\sigma$ is surface tension, $\rho_W$ is the density of water, R is the ideal gas constant and T is temperature. In using Eqn. 2, it is assumed here that $\sigma$ is equal to that of water (72 mN/m). Since the true value of $\sigma$ may deviate from 72 mN/m, we refer to the derived $\kappa$ as the apparent $\kappa$ (or $\kappa_{app}$). The supersaturation

as a function of temperature gradient in the instrument was calibrated with NaCl in the scanned range (Figure S2).

## 2.3   Size and Composition

Diameters of the size-selected, dry, coated particles were measured with a scanning mobility particle sizer (SMPS; consisting of a DMA TSI Model 3081 and TSI CPC 3775). Mode

diameters were obtained by fitting a lognormal function to the obtained size distributions. The organic volume fraction was calculated as:

$$\varepsilon_{org} = \frac{\frac{\pi}{6}D_{tot}^3 - \frac{\pi}{6}D_{NaCl}^3}{\frac{\pi}{6}D_{tot}^3} \tag{3}$$



where $D_{tot}$ is the total coated diameter and $D_{NaCl}$ is the diameter of the size-selected NaCl

particles, adjusted for shape effects. Precision-based uncertainty for the selected diameter was

assessed by size-selecting particles with one DMA and measuring the resulting size distribution.

The mode diameter and the size-selected diameter agreed within 1%. For a subset of

experiments, a scanning electrical mobility sizer (SEMS; BMI, Inc.) was used to measure size

distributions.

       The organic composition was monitored with a chemical ionization mass spectrometer

(CIMS) coupled with a thermal desorption chamber to vaporize the particle-phase organic

coating (McNeill et al., 2008). The CIMS used Cl⁻, as opposed to I⁻, as the reagent ion so as to

retain the ion-molecule adducts (Cl⁻·HA) within the mass range of the quadrupole mass analyzer.

After each experiment, the CIMS sampled particle-free air by setting the second DMA voltage to

0, downstream of the oven, to characterize background gas-phase concentrations. Each

compound had a unique spectrum with an identifiable ion corresponding to the unfragmented

parent compound. The counts for each compound were used to calculate relative abundances for

the binary organic mixtures under the assumption that the sensitivity of the CIMS was the same

for each carboxylic acid. When I⁻ is used as the reagent ion, the CIMS method is somewhat more

sensitive towards palmitic acid compared to oleic acid (Lee et al., 2014). If this difference in

sensitivity similarly applies to the Cl⁻ reagent ion then the relative abundance of palmitic acid

will be underestimated by the assumption of equal sensitivities.

**2.4   Upper-Limit Surface Pressure Calculation**

       The combination of $D_{wet}$ from the CFSTGC, the mode coated diameter ($D_{dry}$), and the

calibrated RH allows for the surface tension ($\sigma$) of the droplets to be calculated for every

individual measurement using Eqn. 1. Since the particles were mixtures of NaCl and organic

compounds, the $\kappa_{avg}$ term was calculated, assuming volume mixing, as:

$$\kappa_{avg} = \varepsilon_{org}\kappa_{org} + (1 - \varepsilon_{org})\kappa_{NaCl} \qquad\qquad (4).$$

The values of $\kappa_{org}$ and $\kappa_{NaCl}$ were assumed to be 0 (Petters et al., 2016) and 1.3 (Petters and

Kreidenweis, 2007), respectively. The use of Eqn. 1 provides a lower limit estimate of $\sigma$, since it

is assumed that none of the organic component partitions into the bulk droplet and is present only

at the surface. These values will therefore be referred to as the lower-limit $\sigma$. To facilitate the

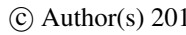


comparison between bulk surface tension studies and this work, the surface pressure ($\pi$) was calculated from the observed $\sigma$ values as:

$$\pi = \sigma_{H2O} - \sigma_{obs} \tag{5}$$

where $\sigma_{H2O}$ and $\sigma_{obs}$ are the surface tension of water and the observed surface tension,

respectively. The $\pi$ values calculated from Eqn. 5 using the lower-limit $\sigma$ will thus be referred to as the upper-limit $\pi$ estimate. A corresponding molecular area ($A$), which is a measure of the inverse of concentration of surfactant at the interface, is calculated as

$$A_{mlc} = \frac{MW}{\rho \cdot N_A}\left(\frac{6 \cdot D_{wet}^2}{D_{coat}^3 - D_{seed}^3}\right) \tag{6}$$

where $MW$ is the molecular weight, $\rho$ is the density of the organic compound, and $N_A$ is

Avagadro's number. For experiments where NaCl was coated with two types of organic compounds, a molar weighted average of the molecular weight and density for each component was used to calculate molecular area.

## 2.5 Surface and Bulk Partitioning

Given the large surface area to volume ratios in microscopic droplets, it is necessary to

account for surface and bulk partitioning when deriving $\sigma$ values from the observations. The compressed film model was used for this purpose (Ruehl et al., 2016). The organic compounds at the air-water interface can contribute to $\sigma$ depression, while organic compounds dissolved into the bulk contribute to droplet growth through the Roault effect. The film model is a 2-dimensional (2D) equation of state (EoS) that parameterizes $\sigma$ as a function of molecular area as:

$$\sigma_{obs} = \min(\sigma_{H2O}, \max(\sigma_{H2O} - (A_0 - A)m_\sigma, \sigma_{min})) \tag{7}$$

where $A_0$ is the critical molecular area, $A$ is the molecular area, $m_\sigma$ is a term that accounts for the interaction between surfactants at the interface, and $\sigma_{min}$ is an imposed lower-limit for $\sigma$. In this model, reductions in $\sigma$ relative to water requires the formation of a full monolayer, which occurs at molecular areas smaller than $A_0$. At molecular areas larger than $A_0$, the droplet $\sigma$ is assumed to

be equal to 72 mN/m and the surface phase is said to be in a "gaseous" state where molecules present at the surface do not interact. Strictly speaking, $\pi$ is non-zero in this state, but we assume this increase in $\pi$ is negligible and use $\sigma = 72$ mN/m. A 2D phase change occurs at $A_0$ and further addition of surfactant molecules to the interface ($A < A_0$) causes compression of the surface film.





This leads to a sharp decrease in σ, or increase in π. The corresponding isotherm for this EoS is given as

$$\ln\left(\frac{C_{bulk}}{C_0}\right) = \frac{(A_0^2 - A^2)m_\sigma N_A}{2RT} \tag{8}$$

where $C_0$ is the bulk concentration at the 2D phase transition, $C_{bulk}$ is the bulk concentration, $R$ is
the gas constant, $N_A$ is Avogadro's number, and $T$ is temperature. The variation of σ with $A$ is
solved for from the observations by minimizing the chi-square value for $D_{wet}$ by varying $A_0$, $m_\sigma$,
and $C_0$ with $D_{coat}$, $D_{NaCl}$, and $RH$ as inputs. The surface tension is constrained by always be larger
than some minimum value (σ$_{min}$ or π$_{max}$) that is determined as part of the data fitting. When the
system reaches σ$_{min}$ the addition of more molecules or increased compression causes dissolution
of surface molecules into the bulk and does not further depress surface tension. An outer iteration
uses the Köhler curve to solve for $D_{wet}$ for a given $RH$. An inner iteration solves Eqn. 8 for the
fraction of organic matter at the surface ($f_{surf}$). For each assumed $f_{surf}$, the inner iteration
calculates a value for $C_{bulk}$ as

$$C_{bulk} = \frac{(1 - f_{surf})(D_{coat}^3 - D_{seed}^3)}{D_{wet}^3 \bar{V}_{org}} \tag{9}$$

and $A$ as

$$A = \frac{6\bar{V}_{org}D_{wet}^2}{f_{surf}(D_{coat}^3 - D_{seed}^3)N_A} \tag{10}.$$

The film model as used here assumes that all organic molecules are either dissolved in the bulk
or located at the droplet surface. The σ is calculated from the $A$ value using Eqn. 7. $D_{wet}$ is then
calculated from the water activity (i.e. the Raoult term) using the number of moles of organic in
the bulk droplet (determined from ε$_{org}$ and $f_{surf}$) and the number of moles of NaCl (determined
from $D_{NaCl}$). The Kelvin term is calculated based on the derived σ value. The use of the film
model results in a single curve that represents the best-fit π-$A$ relationship, rather than the point-
by-point determination that results from the upper-limit method. Uncertainty in the film model
curve was estimated by perturbing the input $RH$ values by the average precision-based
uncertainty in $RH$. The data were then fit to the film model using the perturbed (+ and -) $RH$
values, and the uncertainties were calculated as the difference between the original and perturbed



cases. Some of the differences between the film model results and the upper-limit results is due to the upper-limit coming from a point-by-point analysis while the film model is a fit across all data points. If the upper-limit values are on the whole equal to film model $\pi$ values, this indicates that dissolution is minimal and that droplet growth enhancement is caused enhancements in $\pi$ (or

reductions in $\sigma$). However, if the upper-limit values are systematically lower than the film model $\pi$ values, then droplet growth enhancement is due to both enhancements in $\pi$ and organic compounds dissolving into the bulk.

In addition to the compressed film model, the data were also fit to the Szyszkowski-Langmuir EoS:

$$\sigma_{obs} = \sigma_{H2O} - \frac{RT}{A_0 N_A} \ln\left(1 + \frac{C_{bulk}}{C_0}\right)$$  (9)

where $C_0$ is the bulk concentration at which half of the surface sites are occupied and $A_0$ is the maximum surface concentration. The corresponding isotherm is:

$$\frac{A}{A_0} = \frac{C_{bulk}/C_0}{1 + C_{bulk}/C_0}$$  (10).

In contrast to the film model, the Szyszkowski EoS does not include an interaction parameter.

The Szyszkowski EoS allows for the continuous adsorption and desorption of surfactants from the interface and does not include a 2D phase transition.

## 3    Results and Discussion

### 3.1    Single surfactant systems

#### 3.1.1    Oleic acid coatings

Variation in the observed wet diameters for 200 nm size-selected NaCl particles coated with oleic acid are shown as a function of the coated particle diameter for one RH (~99.94%) in Figure 2A. If it is assumed that $\pi$ is constant and equal to that of water (i.e. $\sigma = 72$ mN m$^{-1}$), the predicted $D_{wet}$ from Eqn. 1 are much lower than the observed $D_{wet}$, with the difference between the two increasing as the coated diameter (and $\varepsilon_{org}$) increases. This observation indicates that the

$\pi$ values in the droplets must be greater than 0 mN m$^{-1}$ ($\sigma_{obs} < \sigma_{H2O}$) for these coated particles, or equivalently that $\sigma$ is lower than that of pure water. The difference between the observations and the assumed pure water curve increases with the dry coated particle diameter (and the $\varepsilon_{org}$), but



only once the coating reaches a critical value. This threshold behavior, or a minimum required $\varepsilon_{org}$ to observe an enhancement in the droplet diameter, was previously observed by Ruehl and Wilson (2014) for ammonium sulfate particles coated with oleic acid and other fatty acids.

The wet droplet diameters are used to calculate upper-limit $\pi$ values from Eqn. 1 for all
NaCl core sizes, $RH$, and $\varepsilon_{org}$ used (see Figure S3), and are considered as a function of molecular area (Figure 2B). Experiments with NaCl coated with 99% purity oleic acid and 90% purity oleic acid were similar, indicating that this change in oleic acid purity did not impact observed $\pi$ values (Figure S4). Notably, these upper-limit estimates of $\pi$, when plotted versus $A$, collapse onto a single curve independent of the selected NaCl seed diameter or measurement $RH$ (99.85-100.03%). This
occurs because when $RH$ is increased (for a given seed diameter), the droplet is larger and the molecular area increases, which then decreases the calculated $\pi$. Similarly, variations in the NaCl seed diameter (with $RH$ held constant) lead to variations in $\varepsilon_{org}$ for a constant coated dry diameter, but this in turn leads to corresponding variations in surface concentrations, molecular areas, and ultimately calculated $\pi$ values. Consequently, the data all collapse onto one curve.

The corresponding compressed film model fit is also shown, both as $D_{wet}$ versus the dry particle diameter (Figure 2A) and as $\pi$ versus molecular area (Figure 2B). Below the threshold $\varepsilon_{org}$ or above the corresponding $A_0$ the film model gives $\pi = 0$. In this low-coverage region the surfactant concentration is below monolayer coverage and the molecules are in a "gaseous" state. Above the threshold $\varepsilon_{org}$ (or below $A_0$), a full monolayer forms and $\pi$ begins to increase. In this
compression region, as more organic molecules are added (i.e. at larger $\varepsilon_{org}$) compression of the monolayer leads initially to a steep increase in $\pi$ as the molecular area decreases. Eventually $\pi$ reaches a maximum and remains constant at this value. This could be indicative of monolayer collapse, the formation of a 3-dimensional phase, and/or dissolution. This region of the $\pi$ isotherm will be referred to as the collapsed region. In contrast, the Szyszkowski EoS does not permit a 2-
dimensional phase transition and thus cannot reproduce the observed droplet growth behavior (Figure 2A). Instead, the Szyszkowski EoS yields wet diameters that increase continuously with the coated diameter and that do not exhibit the plateau at high coverage. This indicates that the Szyszkowski EoS is not appropriate for use with molecules such as oleic acid.

The best fit from the film model for oleic acid on NaCl gives $A_0 = 48.5 \pm 3.8$ Å$^2$, $m_\sigma = 2.15$
$\pm 0.1$ mJ m$^{-2}$, and $\pi_{max} = 37.5 \pm 4.8$ mN/m (see Table 1). In the compression region, the film model





$\pi$ values are close to upper-limit $\pi$ values. This indicates that the effect of bulk partitioning on the Raoult term is small (bulk concentrations < 5%$_{vol}$) and nearly all of the enhanced droplet growth is due to enhancements in $\pi$. In the collapsed region, the film model $\pi$ values are slightly less than the upper-limit estimates, which indicates that bulk partitioning of the organic compounds

becomes important following monolayer collapse.

The $\pi$-$A$ isotherm determined here using the film model for microscopic droplets can be compared to similar measurements for bulk systems (see Figure 2). In the bulk measurements, variation in $A$ is induced through physical compression of the surfactant in a Langmuir-Blodgett trough. The $A_0$ values for the wet droplets compare well to those in bulk systems (Voss et al.,

2007;Mao et al., 2013;Seoane et al., 2000). However, the film model derived $\pi$ values seem to be higher than the bulk system, both in the compression and collapsed region. We hypothesize that this might reflect real physical differences between the droplet and the bulk experiments. For one, in typical bulk experiments, the molecular areas are changed slowly (0.2 to 5 Å$^2$ min$^{-1}$) because they are aimed at measuring under equilibrium conditions. In contrast, in the droplet experiments

water uptake and growth occurs rapidly, in just a few seconds. In addition, the droplet growth experiments proceed from a state where the hygroscopic salt core is covered with a purely organic, initially very thick coating (at the highest $\varepsilon_{org}$). This thick coating is stretched out as water uptake by the inorganic core causes the overall particle to increase in size. This is opposite that in typical Langmuir-Blodgett experiments, which start at an expanded state and compress over time. Some

bulk studies indicate that higher compression rates lead to higher surface pressures during monolayer compression and higher collapsed $\pi$ (Rabinovitch et al., 1960;Jeffers and Daen, 1965;Wüstneck et al., 2005), attributed to time-dependent structural changes in the monolayer. Additionally, bulk studies measuring time-dependent changes in $\pi$ following rapid expansion indicate that relaxation to the equilibrium state can occur with time constants on the order of 10

seconds (Murray and Nelson, 1996) to several minutes (Smith and Berg, 1980), and dynamic surface pressure measurements of atmospheric aerosol extracts indicate that full relaxation can take hundreds of seconds (Noziere et al., 2014).

### 3.1.2   Myristic acid coatings

Upper-limit estimates of $\pi$ and the film model fit to myristic acid coated NaCl particles

are shown in Figure 3. Like oleic acid, no enhancements in $\pi$ are observed above a critical





threshold at $A_0$, i.e. $\pi = 0$ when $A > A_0$. For myristic acid, $A_0 = 29.2 \pm 1.2$ Å$^2$ and is smaller than that for oleic acid. This indicates that myristic acid packs more efficiently than oleic acid and that more surfactant molecules are needed at the surface to enhance $\pi$. We suspect this difference arises because oleic acid has a *cis* double bond, which adopts a bent configuration. This results in less efficient packing at the droplet surface compared to myristic acid, which is a straight chain alkanoic acid (Kanicky and Shah, 2002). The upper-limit $\pi$ estimates are comparable to the film model. This indicates that bulk solubility played a minimal role.

Takahama and Russell (2011) predict from molecular dynamics studies that the mass accommodation coefficient of water on myristic acid coated particles can be suppressed, with the reported range being 0.0-0.04. Here, there is no evidence that such suppression impacted droplet growth, in contrast to experiments with palmitic acid, discussed below. These findings are consistent with Ruehl and Wilson (2014) for ammonium sulfate particles coated with myristic acid. It may be that myristic acid coatings do reduce the accommodation coefficient from unity, but that the ultimate extent of reduction is insufficient to have a substantial impact on the droplet growth on the timescale of these experiments. In general, the experiments here are reasonably insensitive to variations in the accommodation coefficient when greater than 0.01 (Ruehl and Wilson, 2014).

The myristic acid isotherm for droplets determined here is compared to bulk measurements (see Figure 4). Two different isotherms are shown, both taken from Albrecht et al. (1999). One is for myristic acid compressed on a fresh subphase (water), which corresponds to the first in a series of monolayer compression-expansion cycles. The other is for a used subphase, which corresponds to the 8$^{th}$ compression, and where additional surfactant is added prior to each compression. The two isotherms differ substantially. The difference between sequential compression-expansion cycles becomes smaller as the number of cycles increases, and eventually there is little difference between two sequential cycles. Albrecht et al. (1999) concluded that that myristic acid (as well as other short chain fatty acids) slowly dissolves into the subphase, with the effects of this process declining over time as the subphase becomes saturated. Thus, whether a fresh or a reused subphase has a significant impact on the resulting isotherm for bulk systems. This makes quantitative comparison between the myristic acid isotherm from this study and previous bulk studies difficult. The $A_0$ for the isotherm observed



here on the growing droplets is most similar to the fresh subphase ($A_0 \sim 25$ mN m$^{-1}$) and is substantially smaller than the used subphase ($A_0 \sim 50$ mN m$^{-1}$). Dissolution decreases myristic acid molecules from the interface, which means that lower apparent molecular areas (assuming no dissolution) are required to change $\pi$. However, the dissolution of myristic acid into the bulk

is slow, occurring over timescales of 10's of minutes (Albrecht et al., 1999). These time scales are much longer than the time scale for droplet growth (seconds). Based on these time scales, dissolution should not play a large role the experiments presented here, consistent with the similarity between the upper-limit $\pi$ and compressed film model. However, observations from Smith and Berg (1980) indicate that larger $\pi$ can lead to greater rates of dissolution (10%

decrease in molecular area over 3 minutes at $\pi \sim 22$ mN/m). In droplet experiments, the interface is expanding from a highly compressed state (high $\pi$). Therefore, it is possible that dissolution was significant for these experiments. However, this conflicts with film model measurements matching the upper-limit because it is assumed that solubility is negligible (i.e. $\kappa_{org} = 0$) for upper-limit calculations. Nevertheless, the film model is able to reproduce general $\pi$-$A$ behavior

of this system, even if bulk phase concentration is significant.

### 3.1.3 Palmitic acid coatings

Figure 4A shows droplet diameters as a function of coating thickness for NaCl particles coated with pure palmitic acid. In the range of temperatures used in these experiments (< 25 °C), palmitic acid exists as a waxy solid (Inoue et al., 2004), though it is possible that supercooling

can occur (Hearn and Smith, 2005). When palmitic acid is coated on the NaCl particles with $\varepsilon_{org}$ ≥ 0.8, the observed droplet sizes were much lower than what is observed for just the uncoated NaCl core. As coating thickness (and $\varepsilon_{org}$) increases, the extent of suppression in droplet growth decreases. This indicates that the presence of pure palmitic acid as a coating, especially at thick coatings, inhibits droplet growth. Thus, $\pi$ values could not be determined for these experiments.

This behavior is consistent with the low mass accommodation coefficients ($\alpha$) calculated from Ruehl and Wilson (2014) for ammonium sulfate particles coated with palmitic acid. The dependence of droplet growth suppression on coating thickness is likely due to the slow initial diffusion of water through the palmitic acid coating. Thicker coatings apparently lead to greater inhibition of water transport and slower water diffusion to the NaCl core, corresponding to an

initial period of low $\alpha$. However, as some water infiltrates to the salt core and growth due to



water uptake occurs, palmitic acid may form islands (Davies et al., 2013;Lovrić et al., 2016). This may lead to the formation of holes that would enhance water transfer, leading to an increase in the effective α.over time, which allows the droplets to grow to the observed sizes.

### 3.2 Binary surfactant systems

5       To understand how mixing of different organic compounds might impact the ability of compounds such as palmitic acid to inhibit water uptake, experiments were carried out for binary oleic acid-palmitic acid coating. For these experiments, a 1:1 molar mixture of these surfactants was added to the oven. The CIMS composition measurements indicated that the average oleic acid fraction on the particles was $0.33 \pm 0.08$ (Figure S5), slightly lower than that of the mixture

in the oven. This difference likely results from differences in the vapor pressures of these compounds (Cappa et al., 2008), although could result from differences sensitivity of the CIMS to these compounds (Lee et al., 2014). For the mixed palmitic acid and oleic acid coatings on NaCl seeds, the observed droplet sizes were equal to or larger than that of the pure NaCl component, with the droplet size increasing as the total coating thickness increased. This

indicates that, unlike for pure palmitic acid, the mixed coating does not inhibit water uptake. Thus, there is no evidence of kinetic limitations for this system. Presumably, the mixing of liquid oleic acid  (Inoue et al., 2004) with solid/waxy palmitic acid prevented the palmitic acid from packing tightly enough to inhibit initial water uptake.

      Since no kinetic limitations were observed for NaCl particles coated with palmitic acid-oleic

acid mixture, it is possible to determine $\pi$ values for these experiments. The $\pi$-$A$ isotherms from the upper-limit and film models for the oleic acid-palmitic acid mixture (on NaCl seeds) are shown in Figure 4B. These are compared with both the film model curve for oleic acid determined above and observations from bulk palmitic acid measurements (Tang et al., 2010). The film model fit yielded $\pi_{max} = 43.6 \pm 6.3$ mN/m and $A_0 = 35.2 \pm 3.3$ Å$^2$ (Table 1). The $A_0$ is

smaller than that for pure oleic acid coated particles by 10 Å$^2$, indicating that oleic acid packs more efficiently when mixed with palmitic acid. The observed film model isotherm for this system can be compared to the isotherm of an ideal mixture (Adamson and Gast, 1967). The ideal isotherm was calculated from a molar-weighted average of the pure oleic acid isotherm (from the film model fit) and the bulk palmitic acid isotherm (Tang et al., 2010). The observed $\pi$

at a given molecular area was higher than the ideal prediction, closer to oleic acid than to



palmitic acid. This is consistent with behavior observed in previous work for oleic acid-stearic acid mixtures in bulk solutions (Feher et al., 1977) and indicates some difficulty incorporating the bent double bond in oleic acid into the monolayer. (Stearic acid is a saturated fatty acid that is two carbons longer than palmitic acid and has been observed to suppress water uptake,

similarly to palmitic acid (Ruehl and Wilson, 2014).) Other work has found that saturated and unsaturated fatty acids do not mix ideally, but mixed unsaturated fatty acid-saturated fatty acid films instead form distinct domains enriched in either component (Ocko et al., 2002).

The isotherm for palmitic acid from the bulk measurements exhibits more complex behavior than is observed for either oleic acid (either in the bulk or in droplets here) or for the

mixture here. For palmitic acid in bulk experiments, the $\pi$ increases as $A$ decreases through the compression region until it reaches a sharp maximum. At even greater compression (smaller $A$) the $\pi$ then decreases until reaching a plateau. The behavior of palmitic acid stems from the formation of complex 3D structures following monolayer collapse. If the palmitic acid and oleic acid in the mixture were to form distinct domains there should be two distinct collapse pressures

(Ocko et al., 2002;Griffith et al., 2012). Here, the derived upper-limit $\pi$ values for the mixture do not appear to decrease at small $A$ (after the monolayer collapse), instead continuing to rise steadily with decreasing $A$. However, some of this continued increase in $\pi$ likely results from attribution of dissolution of palmitic acid and oleic acid to surface tension in the upper-limit model. In comparison, the $\pi$ values from the film model plateau at small $A$ for the mixture, rather

than peak, decline, and then plateau as with pure palmitic acid, although it should be noted that the film model is not designed to capture such complex behavior.

Previous modelling (Takahama and Russell, 2011) and experimental (Ruehl and Wilson, 2014;Davies et al., 2013) studies, along with our measurements above, indicate that there can be a kinetic limitation to water evaporation and uptake imposed by films composed of single-

component surface active organic species, in particular long-chain organic species that form solid films, which pack densely and exhibit long-range order. However, CCN measurements of ambient particles (Raatikainen et al., 2013) suggest that water uptake and droplet growth are not kinetically limited (i.e. $\alpha > 0.1$) for particles sampled at a variety of locations around the world. Here, the observations demonstrate that mixing of one component that did not inhibit water

uptake (oleic acid) with another that did (palmitic acid) completely removed the kinetic



inhibition. Most likely, the mixing of the two components prevented the formation of a tightly packed film and so water uptake was facile. These observations serve as a potential explanation for why low α (< 0.1) water values are not observed for particles in the ambient atmosphere (Raatikainen et al., 2013), since ambient particles are multi-component mixtures. They also

support the suggestion by Davies et al. (2013) that kinetic inhibition to water uptake should rarely be observed in ambient particles, although it is possible that kinetic limitations could become more pronounced at lower temperatures, as decreasing temperature leads to increasing packing density (lower molecular area) (Davies et al., 2013).

     Measurements were also made for a mixture of myristic acid and oleic acid coated on

NaCl particles. For this mixture, the coating composition on the particles was dominated by myristic acid (~90% by mole), even though the mixture composition in the coating apparatus was 1:1 by mole. The upper-limit π estimates as a function of molecular area are shown in Figure S6. At molecular areas > 10 Å$^2$, the π estimates for the mixture approximately match the pure myristic acid case, but for molecular areas <10 Å$^2$, the surface pressures for the mixed case were

slightly larger. The general consistency between the mixed and pure case is consistent with the high myristic acid fraction.

### 3.3  Köhler curves and the surface tension at activation

     Above, we showed that the addition of sufficient amounts of fatty acid surfactants to salt particles (here, NaCl) can produce substantial σ depression (or π enhancement) in droplets near

the point of activation, that is just above or below 100% RH. These observations are consistent with other similar observations for both fatty acids and other compounds (Ruehl et al., 2012;Ruehl et al., 2016;Ruehl and Wilson, 2014). However, an important question is the extent to which this σ depression ultimately impacts activation into cloud droplets. Here, this is assessed by measuring CCN activation curves for NaCl particles coated with: oleic acid; oleic

acid mixed with either palmitic acid or myristic acid; and for oleic acid particles oxidized by O$_3$. These activation curves were used to calculate the apparent κ values from the measured critical supersaturation ($s_c$). The observed $κ_{app}$ values are compared with those predicted assuming volume mixing ($κ_{mix}$; Eqn. 4). Any enhancement in $κ_{app}$ values relative to the volume mixing line is attributed to σ depression. The observed $κ_{app}$ values are generally consistent with volume

mixing rules, although there are some small differences at higher $ε_{org}$ (Figure 5A). Our





observations are consistent with Nguyen et al. (2017), who found good agreement between observed and calculated $\kappa_{app}$ values for sea salt particles coated with various long-chain unsaturated fatty acids, including oleic acid. Values of σ at activation reported here were estimated based on the small differences between the $\kappa_{app}$ and the $\kappa_{mix}$ by calculating the σ

required to achieve perfect closure between the observed and calculated $s_c$ when it is assumed that $\kappa = \kappa_{mix}$. For $\varepsilon_{org} < 0.90$, the calculated values were very close to 72 mN/m. This indicates that the surfactants have little influence on the CCN activation (i.e. on the $s_c$) despite there being substantial depression of σ at RH values slightly lower than the critical value for similar $\varepsilon_{org}$. Above $\varepsilon_{org} > 0.90$, the average calculated σ was $66.4 \pm 2.9$ mN/m, slightly less than that for pure

water. This indicates that when these surfactants are sufficiently abundant there is a small, but non-negligible influence of σ depression on activation.

       To understand in greater detail the influence of these types of surfactants on CCN activation and to examine the robustness of the conclusion that the surfactants have some influence on activation at sufficiently high $\varepsilon_{org}$, experiments were performed in which the

composition of the particle was held constant ($\varepsilon_{org} \sim 0.95$, using oleic acid) while RH was varied from just below 100% RH to the point of activation. Figure 5B shows the directly observed Köhler curve for these particles, that is the variation in $D_{wet}$ with RH. The individual points are colored according to the derived σ calculated from the upper-limit estimates (Eqn. 1). As RH is increased and the particles approach activation, the droplets grow. The growth leads to an

increase in both the droplet surface area and the corresponding molecular area, since the number of organic molecules is fixed. This leads to a fairly continuous increase in σ over the RH range considered. Upon sufficient growth the molecular area increases beyond $A_0$ and the reduction in σ becomes zero. This underscores the importance of considering surface-area normalized concentrations as opposed to bulk concentrations.

Importantly, the $D_{wet}$ at which the observed Köhler curve intersects the constant-κ Köhler curve (calculated assuming σ remains constant at 72 mN m$^{-1}$) is past the constant-κ maximum (Figure 5B). Consequently, the observed $s_c$ (= 0.057%) is slightly lower than that predicted value assuming constant σ ($s_c = 0.062\%$), consistent with the comparison between $\kappa_{app}$ and $\kappa_{mix}$ above. This indicates that σ depression has some impact on activation, but that the impact on $s_c$ is

relatively small (8%) for the model surfactants used in these experiments. Considered along with



the CCN activation measurements (Figure 5A), it is also apparent that the surfactants impact the critical supersaturation only at particularly high $\varepsilon_{org}$. These modest reductions in $\sigma$ are broadly consistent with results found in Schwier et al. (2011) for NaCl mixed with acidified sodium oleate (where almost all sodium oleate is present as oleic acid) and with Nguyen et al. (2017).

Although the ultimate effects on $s_c$ are small for these surfactants, depressions in $\sigma$ change the trajectory of activation through enhancement of droplet sizes, especially at lower RH (~99.9%), which is consistent with Ruehl et al. (2016). To have a significant impact on activation, organic compounds must substantially reduce the (maximum) critical supersaturation relative to the pure, uncoated salt.

**3.4  Oxidation experiments**

To understand how oxidation affects $\pi$ (or $\sigma$), experiments were conducted in which NaCl particles coated with oleic acid were oxidized with $O_3$ in a flow tube. Particle size and coating thickness were kept constant, while the residence time in the flow tube was varied to change the extent of oxidation. The composition of the oxidized particles was characterized by a CIMS

(Figure 6A). The average spectra for the most oxidized case and for the pure non-oxidized oleic acid are shown in Figure S7. The major oxidation products measured in this study include nonanoic acid, azelaic acid, and 9-oxononanoic acid. As the $O_3$ residence time increased, the absolute abundance and fractional contribution of oleic acid decreased. The most abundant product was 9-oxononanoid acid (~86%), with minor contributions from azelaic acid (~7%) and

nonanoic acid (~7%). The fractional contribution of products is generally consistent with previous work (Katrib et al., 2004;Hearn and Smith, 2004). Note that some oxidation occurred for the zero second case because of the flow tube geometry described above. The diameters of the coated particles decreased relative to the bypass channel upon reaction with $O_3$, likely due to the formation and subsequent evaporation of nonanal. The volume of the particles was reduced

by 8% for lightly oxidized particles and 18% for highly oxidized particles. The $\varepsilon_{org}$ for the oxidized particles ranged from 0.86 to 0.88, after reaction.

The CCN-derived $\kappa_{app}$ values were used to assess changes in solubility of the coating material, assuming that the $\sigma$ of oleic acid coated NaCl is close to that of water at activation, a reasonable assumption given the results shown above. A difference between the observed $\kappa_{app}$

values and the calculated $\kappa_{mix}$ could indicate an increase in the solubility of the organic



component after oxidation. Shown in Figure 5A are the measured $\kappa_{app}$ values for each oxidation condition at the measurement $\varepsilon_{org}$ and the theoretical line for the volume mixing assumption (assuming $\sigma = 72$ mN/m). The measured $\kappa_{app}$ values for the oxidized particles are very similar to both the volume mixing line and to the unoxidized oleic acid coated NaCl particles. This

indicates that changes in solubility were minimal. This minimal change in solubility is supported by the low $\kappa$ values (~<0.003) for 9-oxononoic acid and nonanoic acid predicted from a functional group model (Petters et al., 2016). Azelaic acid has a higher observed $\kappa_{app}$ value of 0.02 (Kuwata et al., 2013), but since this product is only a minor component of the coating material, it does not have a strong impact on the overall $\kappa$.

Given that the observations indicate no substantial change in solubility, upper-limit $\pi$ values for oxidized particles have been determined, as measured before activation at RH ~100%. The $\pi$ values for oxidized particles at a given $A$ are compared to the $\pi$-$A$ isotherm measured for unoxidized oleic acid particles (Figure 6B). Molecular areas were calculated using molar-weighted fractions of oleic acid and the oxidation products. After oxidation, the $\pi$ values of these

particles are still well above that of pure water. There is some indication that the $\pi$ for oxidized particles are lower than that for unoxidized particles at the same molecular area, although the difference is generally small. These observations are reasonably consistent with previous findings in Schwier et al. (2011) for particles composed of acidified sodium oleate and NaCl reacted with $O_3$ at somewhat higher exposures than here (exposure ~8.8 x $10^{14}$ to 4.4 x $10^{15}$

molecules cm$^{-3}$ s). However, unlike the observations presented here, the $\sigma$ for their experiments were estimated at activation based on the observed $s_c$ and so the overall influence of reductions in $\sigma$ should be much smaller.

## 4    Developing broader understanding via the film model

### 4.1    Sensitivity to film model parameters

To go beyond the specific chemical systems experimentally investigated here, the film model can be used. The film model allows for theoretical exploration of the relationship between the properties of surface-active organic molecules (characterized by $A_0$, $m_\sigma$, $C_0$ and $\sigma_{min}$, and also the molecular volume or $v_{org}$) and $s_c$ and $\kappa_{app}$. This serves to establish under what conditions the addition of organic material should lead to a reduction in the $s_c$ (and increase in $\kappa_{app}$) and more





efficient CCN activation, relative to that expected from volume mixing rules. As a starting point, theoretical Köhler curves have been calculated for three specific sets of film model parameters: (i) those determined here for oleic acid, and those determined by Ruehl et al. (2016) for (ii) glutaric and (iii) pimelic acid (Table 1). Glutaric acid and pimelic acid were selected because,

unlike oleic acid, they were observed by Ruehl et al. (2016) to have a substantial impact on $s_c$ and $\kappa_{app}$ through reductions in $\sigma$. Glutaric acid is a 5-carbon and pimelic acid a 7-carbon straight-chain dicarboxylic acid. The calculations use 80 nm NaCl seed particles coated to $\varepsilon_{org} = 0.80$, which corresponds to a coated particle diameter of 136.8 nm (Figure 7A). The selected $\varepsilon_{org}$ value is similar to that observed for SSA with $D_p < 200$ nm generated from laboratory breaking

waves in seawater during a microcosm (Deane et al., Submitted) and to that observed for ambient marine particles in high chlorophyll-a environments (O'Dowd et al., 2004a). The film model results are compared to the Köhler curve calculated assuming the organic component was insoluble and that $\sigma = 72$ mN m$^{-1}$, referred to as the constant $\sigma$ or no $\sigma$ reduction case. All $\kappa_{app}$ values are calculated from the model-predicted $s_c$ values, which account for $\sigma$ reduction,

assuming that there is no $\sigma$ reduction.

For the constant $\sigma$ case the calculated $s_c = 0.142\%$, corresponding to $\kappa_{app} = 0.27$, for these conditions. For oleic acid, the calculated critical supersaturation and $\kappa_{app}$ were unchanged from the constant $\sigma$ case (Figure 7A). This is because the calculated $\sigma$ in the film model reaches 72 mN m$^{-1}$ at a $D_{wet}$ that is substantially smaller than the critical diameter for the constant $\sigma$ case

(Figure 7B). In contrast, the $s_c$ for glutaric and pimelic acid coated particles are reduced slightly and the $\kappa_{app}$ increased slightly, to $s_c = 0.137$ ($\kappa_{app} = 0.29$) and $0.133$ ($\kappa_{app} = 0.31$), respectively. This is because the reduction in $\sigma$ persists to beyond the constant $\sigma$ case activation point for these cases. The particles ultimately activate at the point where $\sigma$ reaches 72 mN m$^{-1}$, the constant $\sigma$ value, as was noted by Ruehl et al. (2016). The difference between oleic acid and the two

dicarboxylic acids results in part from oleic acid having a larger $v_{org}$. Consequently, the molecular area for oleic acid (at $\varepsilon_{org} = 0.8$) substantially exceeds $A_0$ prior to the constant $\sigma$ activation point and there is no impact of the reduction in $\sigma$ observed at slightly lower RH on the actual $s_c$ or $\kappa_{app}$. This is not the case for the two diacids, and thus surface tension depression impacts $s_c$. For comparison, at the activation diameter of the constant $\sigma$ case (0.98 µm) the $A$ for

oleic acid is ~150 Å$^2$ but only 95 Å$^2$ for glutaric acid and 62 Å$^2$ for pimelic acid (see Figure 7C).



Given the above, the general sensitivity of $\kappa_{app}$ to the surfactant properties (and their influence on $\sigma$ depression) has been more systematically assessed, again using 80 nm NaCl seed particles. The predicted $\kappa_{app}$ values from the film model were found to be most sensitive to variations in the $v_{org}$ and the $A_0$ of the organic species (Figure 8). The predicted $\kappa_{app}$ values were

not sensitive to variations in $m_\sigma$ values above 0.1 mJ m$^{-2}$ and variations in $C_0$ had limited impact for $A_0 < 100$ Å$^2$. Additionally, the predicted $\kappa_{app}$ were not sensitive to $\sigma_{min}$ for values below 65 mN/m. Thus, we focus on $v_{org}$ and $A_0$.

The dependence of $\kappa_{app}$ on the organic species $v_{org}$ and $A_0$ is examined for two cases: $\varepsilon_{org} = 0.8$ (Figure 8A) and $\varepsilon_{org} = 0.9$ (Figure 8B). (For these calculations, $m_\sigma = 1.0$ mJ m$^{-2}$, $C_0 = 1$ x $10^{-6}$

mol mol$^{-1}$ and $\sigma_{min} = 40$ mN/m.) For a given $\varepsilon_{org}$, species with smaller $v_{org}$ have smaller molecular areas (higher surface area-normalized concentrations) at a given $D_{wet}$. Thus, the extent of $\sigma$ depression is greater for smaller $v_{org}$. This is consistent with the observations in Ruehl and Wilson (2014). We find that the sensitivity of $\kappa_{app}$ to a change in $v_{org}$ increases as $v_{org}$ becomes small. Likewise, species with larger $A_0$ will require less surfactant at the surface to reduce $\sigma$ and

impact activation. The calculated $\kappa_{app}$ are approximately constant over a wide range of $A_0$ and $v_{org}$, in particular when $A_0$ is small and $v_{org}$ is large, with $\kappa_{app} \sim 0.3$ when $\varepsilon_{org} = 0.8$ and $\kappa_{app} \sim 0.17$ when $\varepsilon_{org} = 0.9$ (Figure 8A). These values are very similar to those predicted from volume mixing rules assuming $\sigma = 72$ mN/m. However, when $A_0$ is instead relatively large and/or $v_{org}$ is relatively small the $\kappa_{app}$ can be very large, with substantial changes in $\kappa_{app}$ predicted for relatively

modest changes in $v_{org}$. The exact transition from the nearly constant $\kappa_{app}$ to the $A_0$ and $\varpi_{org}$-sensitive $k_{app}$ depends on $\varepsilon_{org}$. In general, for larger $\varepsilon_{org}$ the $k_{app}$ becomes sensitive to variations in $v_{org}$ and $A_0$ at larger absolute $v_{org}$. These calculations demonstrate that it is, at least in theory, possible to observe large $\kappa_{app}$ values for mixed inorganic-organic particles even when the organic fraction is large. However, substantial increases in $\kappa_{app}$ are only obtained for a particular range of

organic properties, within the film model framework.

## 4.2  Linking to sea spray and secondary marine aerosol

The chemical systems considered in this study—long-chain fatty acids coated on NaCl—were considered in part because even-numbered fatty acids have been identified as substantial components in submicron sea spray aerosol particles (Cochran et al., 2016;Schmitt-Kopplin et





al., 2012;Mochida et al., 2007). Our measurements demonstrate that the fatty acids can have a substantial impact on σ near activation, when present at sufficient abundance, but that reductions in σ ultimately have limited impact on activation (i.e. on $s_c$). A lowering of $s_c$ due to reductions in σ has been used (or hypothesized) to explain the often large CCN activation efficiency for

particles observed in the marine environment, in particular nascent SSA particles. For example, Ovadnevaite et al. (2011) observed high CCN activity concurrent with low water uptake at subsaturated RH for ambient SSA particles sampled during a field study. In general, at RH values well below 100% the Raoult, or solubility, effect primarily controls water uptake, with limited influence of σ (and thus limited sensitivity to variations in σ). Thus, the low water uptake

under sub-saturated conditions indicates the particles have limited salt content, and the CCN/sub-saturated difference implicates σ as an important factor. As another example, Collins et al. (2016) observed persistently high CCN activation efficiency ($\kappa_{app} > 0.7$) for microcosm studies of nascent SSA with $D_p < 200$ nm regardless of the biological activity within the seawater used. This was even though the SSA with $D_p < 200$ nm particles were likely highly enriched in organic

matter during these mesocosms (Deane et al., Submitted). This again implies an important role for σ in affecting the CCN activity. If σ depression is responsible for the observed high CCN activity of SSA then the average properties of the marine surfactants must differ substantially from the fatty acids systems tested here, given the distinct lack of impact of the fatty acids on $s_c$ observed here and by (Nguyen et al., 2017). Our theoretical analysis above using the film model

suggests that the complex mixture of marine organic compounds (which includes fatty acids) with salts must interact to have an effective $A_0 > 100$ Å$^2$ and overall relatively small $v_{org}$ values.

In Section 4.1, the dependence of $\kappa_{app}$ on $A_0$ and $v_{org}$ was examined for particles with constant $\varepsilon_{org}$. Here, drawing on the Collins et al. (2016) observations, calculations of $\kappa_{app}$ have been performed as a function of $\varepsilon_{org}$ for select pairs of $A_0$ and $v_{org}$. This provides insight into how

variation in the relative abundances of salts and organics in (theoretical) SSA particles could impact $\kappa_{app}$. Three pairs of $A_0$ and $v_{org}$ values were chosen to produce $\kappa_{app}$ values at $\varepsilon_{org} = 0.8$ of $\kappa_{app} = 0.35, 0.50,$ and $0.70$ for an 80 nm NaCl seed particle, which can be compared to $\kappa_{app} = 0.27$ if σ = 72 mN/m. There is more than one ($A_0$, $v_{org}$) pair that yields the same $\kappa_{app}$ (Figure 8A). Thus, the specific pair chosen here were selected to test the sensitivity of $\kappa_{app}$ to each parameter,

and the three pairs considered were (i) $A_0 = 150$ Å$^2$, $v_{org} = 0.6$ x $10^{-4}$ m$^3$ mol$^{-1}$, (ii) $A_0 = 100$ Å$^2$,



$\nu_{org}$ = 0.6 x $10^{-4}$ $m^3$ $mol^{-1}$, and (iii) $A_0$ = 100 $\text{Å}^2$, $\nu_{org}$ = 10 x $10^{-5}$ $m^3$ $mol^{-1}$. For the parameter combinations tested, the $\kappa_{app}$ decreases monotonically with $\varepsilon_{org}$ when $\varepsilon_{org}$ < 0.5, following the volume mixing line assuming constant $\sigma$ = 72 mN/m (Figure 9). The decrease in $\kappa_{app}$ with $\varepsilon_{org}$ in this range results from the increased fraction of insoluble organic material, but with no influence on the $\sigma$ because the organic is not sufficiently abundant. However, at $\varepsilon_{org}$ > 0.5, the organic is sufficiently abundant to reduce $\sigma$ and this buffers the decline in $\kappa_{app}$ that results from addition of insoluble material. That is, at $\varepsilon_{org}$ > 0.5 the calculated $\kappa_{app}$ are relatively insensitive to variations in $\varepsilon_{org}$ and are larger than that obtained assuming $\sigma$ = 72 mN/m. Consistent with the above analysis (Figure 8A), for a given $A_0$ a decrease in $\nu_{org}$ leads to an increase in $\kappa_{app}$, here from 0.35 ($\nu_{org}$ = 1 x $10^{-4}$ $m^3$ $mol^{-1}$) to 0.50 ($\nu_{org}$ = 6 x $10^{-5}$ $m^3$ $mol^{-1}$) when $A_0$ = 100 $\text{Å}^2$ and $\varepsilon_{org}$ = 0.8. Correspondingly, at a given $\nu_{org}$ an increase in $A_0$ leads to an increase in $\kappa_{app}$, here from 0.50 ($A_0$ = 100 $\text{Å}^2$) to 0.67 ($A_0$ = 150 $\text{Å}^2$) when $\nu_{org}$ = 6 x $10^{-5}$ $m^3$ $mol^{-1}$ and $\varepsilon_{org}$ = 0.8. The exact dependence of $\kappa_{app}$ on $\varepsilon_{org}$ depends on the assumed film model parameters (Figure 9). In certain cases, the $\kappa_{app}$ can even increase with increasing $\varepsilon_{org}$. Regardless of the exact behavior, this exercise demonstrates that $\sigma$ depression by mostly insoluble species has the potential to buffer observations of $\kappa_{app}$ against changes in composition. Relevant to nascent SSA particles, in particular, we find it is theoretically possible for this buffering effect to maintain $\kappa_{app}$ > 0.7 even at high $\varepsilon_{org}$. It may be that such buffering effects explain the observations of Collins et al. (2016), who observed variability in the $\kappa_{app}$ values for nascent SSA particles over the range $\kappa_{app}$ = 0.7 – 1.3 for a wide range of chlorophyll-a concentrations, but did not observe $\kappa_{app}$ < 0.7.

Beyond SSA, recent observations of the CCN activity of very small, ambient secondary marine aerosol (SMA) particles have also been interpreted as indicating that $\sigma$ depression has a substantial impact on $s_c$ (Ovadnevaite et al., 2017). Ovadnevaite et al. (2017) report the relationship between $\sigma$ and $D_{wet}$ obtained from thermodynamic calculations of the droplet phase behavior (in particular, liquid-liquid phase separation) and an assumption that $\sigma$ can be calculated as the surface-area-weighted mean of the composition-dependent $\sigma$ for each of the two liquid phases. Here, we have translated the reported $\sigma$-$D_{wet}$ relationship to $\pi$ versus $A$ (see Figure 7C). It was assumed here that the organic mass fraction = 0.55 (as reported) and that $\rho_{org}$ = 1.6 g $cm^{-3}$ and $MW_{org}$ = 332 g $mol^{-1}$; these are the estimated density and molecular weight of the dimer species used as surrogate compounds used by Ovadnevaite et al. (2017), and correspond to a $\nu_{org}$



of 2.1 x $10^{-4}$ m$^3$ mol. The inferred $A_0$ is 1000 Å$^2$. This is much larger than the compounds considered here, and might generally be considered as very large. Consistent with our above general analysis, it is evident that large $A_0$, in addition to sufficiently low $v_{org}$, are necessary to significantly reduce critical supersaturations. The reason for the particularly large derived $A_0$ (based on their reported $\sigma$-$D_{wet}$ relationship) is not totally clear, but is likely related to their assumption of the minimum coating amount (reported as a minimum thickness) necessary to impact $\sigma$. If the minimum thickness for a given compound is low, the buffering effect described above would be significant, even at lower $\varepsilon_{org}$ values, because a much lower concentration of surfactants at the surface (higher molecular area) would be required to reduce $\sigma$.

## 5   Conclusion

In this study, surface tension values were estimated for droplets at RH near activation that were grown from NaCl particles coated with the fatty acids oleic acid, myristic acid or palmitic acid, or a palmitic acid-oleic acid mixture. The retrieved $\sigma$ depended explicitly on the relative amount of organic coating, the exact RH, and the identity of the fatty acid. For particles with $\varepsilon_{org}$ > 0.80, the observed $\sigma$ were reduced significantly compared to pure water. Observed variability in the relationships between $\sigma$, RH and $\varepsilon_{org}$ can be explained by the dependence of the molecular area of the organic molecules on these parameters. When s, or equivalently the surface pressure, is considered as a function of molecular area (i.e. in $\pi$-molecular area isotherms), organic molecule specific relationships are obtained that are independent of whether RH or $\varepsilon_{org}$ is responsible for the variation in the molecular area. The $\pi$-molecular area isotherms from the droplets are used to determine critical molecular areas for each fatty acid. The $A_0$ value for oleic acid on droplets compared well to Langmuir trough measurements on bulk solutions, but the magnitude of $\pi$ values at high compression (i.e. small molecular areas) may be larger in the droplets. For myristic acid, the $A_0$ value on droplets compared best with bulk experiments in which dissolution was significant, i.e. when a fresh subphase was used.

When NaCl was coated with pure palmitic acid there was substantial suppression in water uptake observed. This was likely due to the formation of densely packed films through which water molecules could not efficiently permeate. The suppression disappeared when palmitic acid was mixed with oleic acid, indicating a decrease in packing density. For oleic acid coated NaCl



particles exposed to $O_3$ the $\sigma$ values remain significantly lower than water at large $\varepsilon_{org}$, but may be slightly higher than pure oleic acid coatings. Overall, chemical changes due to oxidation of oleic acid by $O_3$ had minimal impact on the $\sigma$ depression.

Values of the critical supersaturation were also quantified by a CCN counter for comparison to the observations made at RH values just below activation. Despite the large reductions in $\sigma$ observed at RH values just below $s_c$, the measured $s_c$ indicate that the fatty acids have minimal impact on the ultimate activation into cloud droplets. This is because the additional growth as the RH increases to $s_c$ causes the molecular area to rapidly increase above $A_0$, limiting the impact on activation. However, $\sigma$ has a large effect on the trajectory of activation by enhancing droplet sizes when RH is $< s_c$.

The film model of Ruehl et al. (2016) was used to theoretically explore what properties surface active organic compounds must have to have a substantial impact on CCN activation, and not just on the $\sigma$ at RH values very close to (yet below) $s_c$. We find that it is theoretically possible for surface active organics to have a substantial impact on CCN activation efficiency, even though this was not the case for the fatty acids here. In particular, the model $s_c$, and consequently $\kappa_{app}$, are most sensitive to variations in the organic compound molecular volume ($\nu_{org}$) and $A_0$, for NaCl-coated particles with relatively large organic fractions. Further, we show that surface tension depression from surface active organic molecules can serve to buffer $\kappa_{app}$ against changes in the organic-to-salt ratio when the $\varepsilon_{org} > 0.5$. Overall, we conclude that surface active organic molecules can have a substantial impact on CCN activation efficiency. However, the extent to which this will occur is strongly dependent upon the specific molecular properties of the organic molecules, and traditional surfactants (such as fatty acids) can actually have a negligible impact on CCN activation.

## 6    Acknowledgements

This study was funded by the Center for Aerosol Impacts on Climate and Environment (CAICE), a NSF Center for Chemical Innovation (CHE-1305427). The authors also thank Hansol Lee and Alexei Tivanski (University of Iowa) and Michael Vermeuel (UW Madison) for their input and support for this project.





## 7 Data Availability

All data associated with figures in this manuscript are archived at the DASH digital archive as

part of the California Digital Library with DOI: 10.25338/B84K5G.

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



**Table 1.** Summary of experiments and corresponding compressed film model fit parameters. Uncertainties on the film model fit parameters on the precision in RH (see text for details).

| Coating | Oleic Acid Fraction | $A_0$ ($\text{Å}^2$) | $m_\sigma$ (mJ m$^{-2}$) | log $C_0$ (mol mol$^{-1}$) | $\pi_{max}$ (mN/m) |
|---|---|---|---|---|---|
| Oleic Acid | 1 | $48.5 \pm 3.8$ | $2.15 \pm 0.10$ | $-6.10 \pm 0.38$ | $37.5 \pm 4.8$ |
| Myristic Acid | 1 | $29.2 \pm 1.2$ | $1.28 \pm 0.20$ | $-7.40 \pm 0.08$ | ** |
| Oleic Acid and Palmitic Acid | $0.33 \pm 0.08$ | $35.2 \pm 3.3$ | $5.41 \pm 0.88$ | $-5.24 \pm 0.22$ | $43.6 \pm 6.3$ |
| Oxidized Oleic Acid[#] | Variable | # | | | |
| Glutaric Acid[&] | | 62 | 1.06 | -7 | 42 |
| Pimelic Acid[&] | | 95 | 0.66 | -6 | 26 |

[#] These experiments were conducted only at one $\varepsilon_{org}$ and thus the data are too limited for application of the film model

** No minimum surface tension was reached for this experiment

[&] From Ruehl et al. (2016)





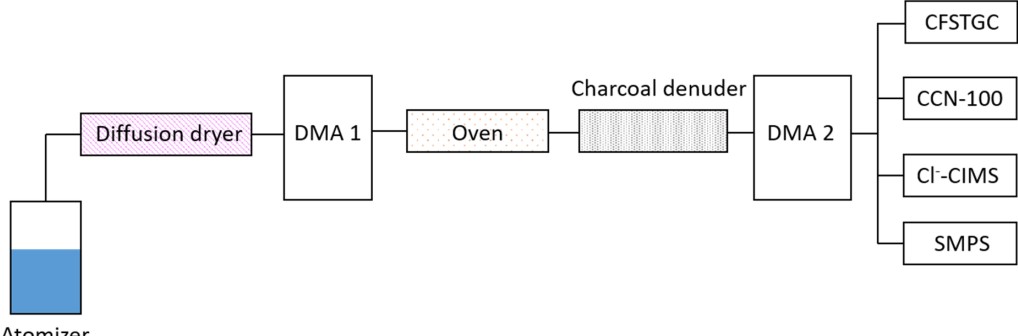

**Figure 1.** Experimental schematic.



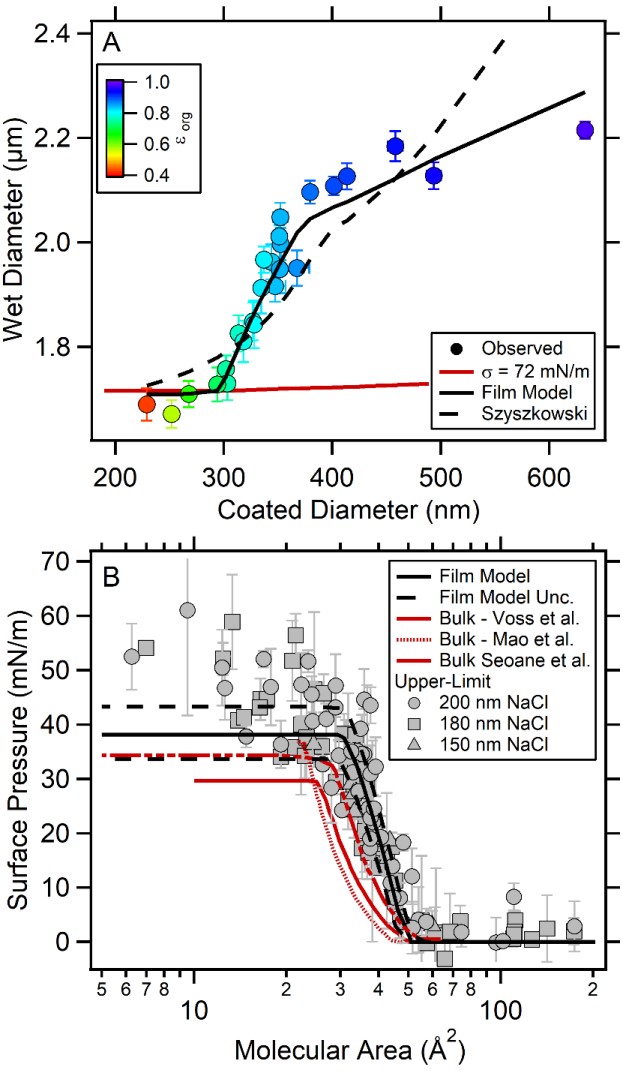

**Figure 2. (A)** Wet diameter (at RH = 99.93%) vs. dry diameter of 200 nm NaCl particles coated with varying amounts of oleic acid. The red line is the calculated droplet size, assuming σ = 72 mN/m independent of coating amount and the black and dashed black lines are film model and Szyszkowski model fits to the data, respectively. The error bars correspond to the confidence interval for mode of the wet diameter distribution. The data points are colored by organic volume fraction ($\varepsilon_{org}$). **(B)** Upper-limit estimates (grey points) and film model estimates (black lines) for surface pressure as a function of molecular area for various NaCl seed sizes (circles = 200 nm, squares = 180 nm, and triangles = 150 nm) and RH. Error bars on individual points and the film model fit (dashed black lines) are based on the precision in RH. Bulk measurements (red lines) adapted from Voss et al. 2007, Mao et al. 2013, and Seoane et al. 2000 are included for reference.




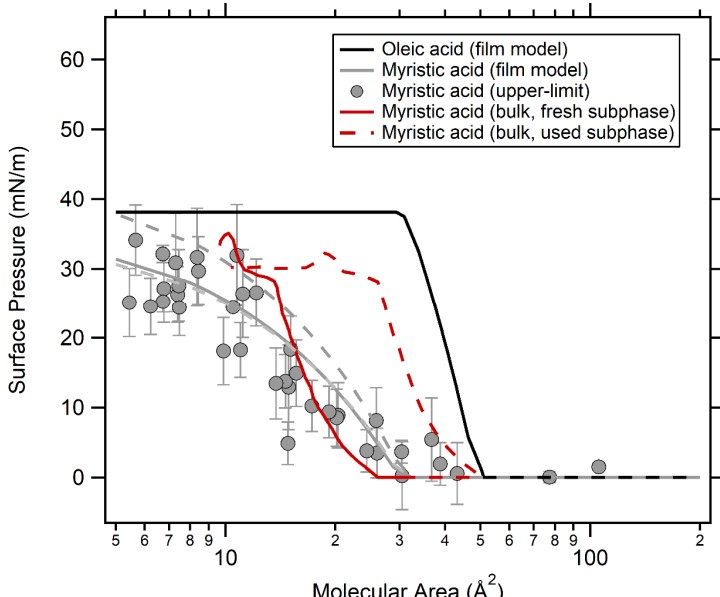

**Figure 3.** Upper-limit and the film model π estimates for NaCl particles coated with myristic
acid as a function of molecular area. Uncertainties in surface pressure are estimated from
precision in RH. The black line is the film model estimate for pure oleic acid and the red line are
bulk measurements of surface pressure adapted from Albrecht et al. (1999).





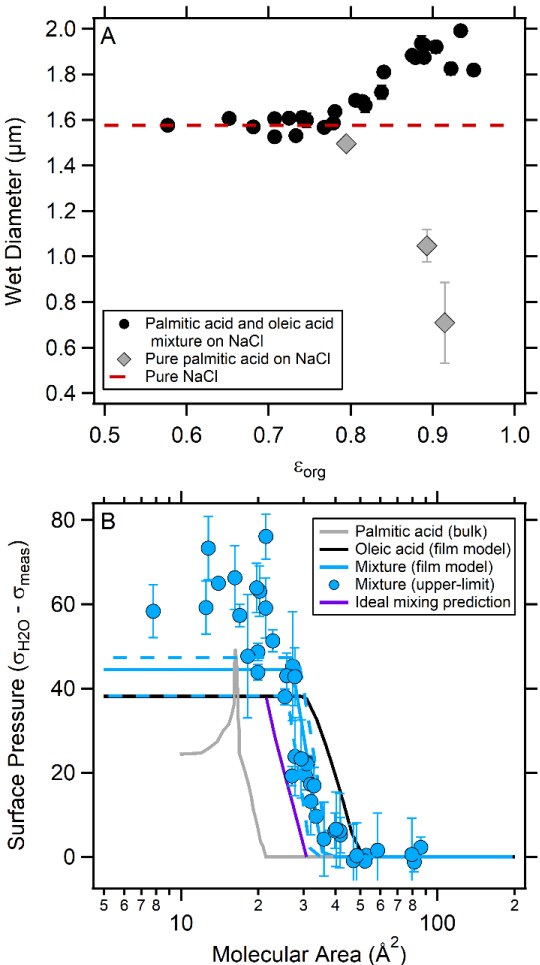

**Figure 4. (A)** Measured wet diameter at RH = 99.88% as a function of organic volume fraction for 200 nm NaCl particles coated with pure palmitic acid (diamonds) and a mixture of palmitic acid and oleic acid (triangles). **(B)** Upper-limit and film model estimates for NaCl particles coated with a mixture of oleic acid and palmitic acid surface pressure as a function of molecular area. The black line is the oleic acid film model estimate and the grey line corresponds to bulk surface pressure measurements from Tang et al. 2010, since surface pressure measurements for NaCl particles coated pure palmitic acid were not possible (see text for details). The purple line is the ideal mixing estimate for surface pressure based on the measured molar fractions of oleic acid and palmitic acid.





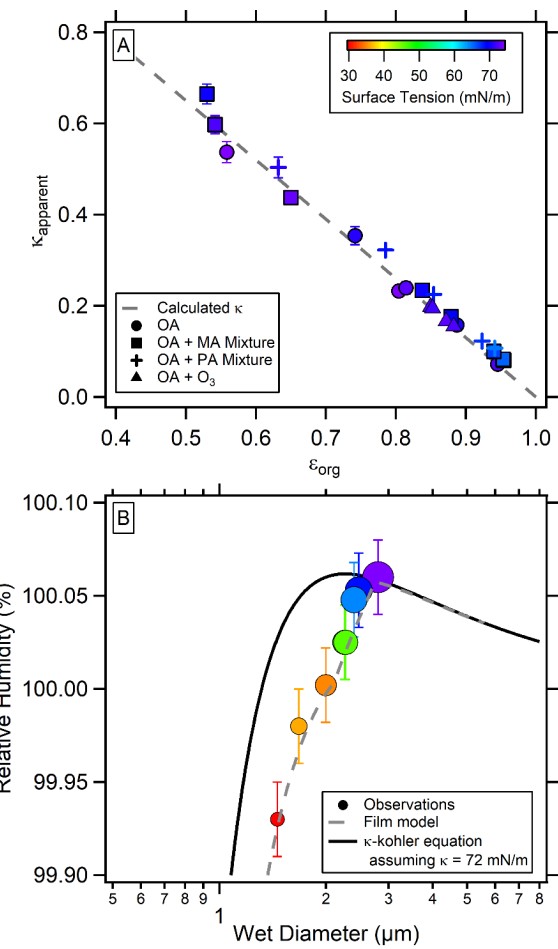

**Figure 5. (A)** Apparent $\kappa$ values as a function of $\varepsilon_{org}$ as calculated from the critical supersaturation for NaCl particles coated with oleic acid (circles), a mixture of myristic acid and oleic acid (squares), a mixture of palmitic acid and oleic acid (crosses) and oxidized oleic acid (triangles). Also shown is the predicted $\kappa$ based on volume mixing rules assuming that $\kappa_{NaCl}$ = 1.3 and $\kappa_{org}$ = 0.001, with $\sigma$ = 72 mN/m. Each point is colored by the actual surface tension required to match observations in $\kappa$. **(B)** Relative humidity as a function of measured wet diameter for 180 nm NaCl particles coated with a fixed amount of oleic acid ($\varepsilon_{org}$ = 0.95). Points are colored by upper-limit estimates for surface tension and the size of the points corresponds to the wet diameter. Also shown are values predicted from the compressed film model and kappa-kohler theory assuming $\sigma$ = 72 mN/m.



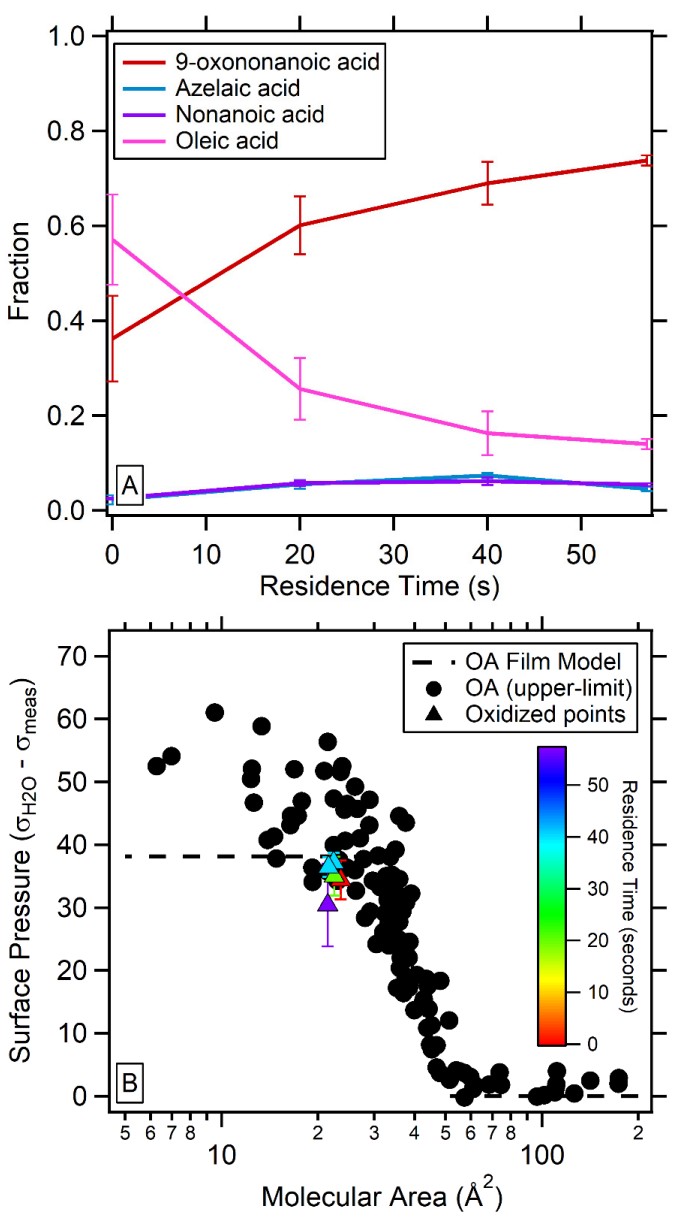

**Figure 6. (A)** Fractional contribution of oleic acid and oxidation products as a function of residence time. **(B)** Surface pressures as a function of molecular area for oxidized oleic acid coated NaCl particles colored by residence time (triangles). Upper-limit surface pressure estimates (black circles) and film model estimate (dashed black line) for unoxidized oleic acid coated NaCl experiments are provided for comparison.



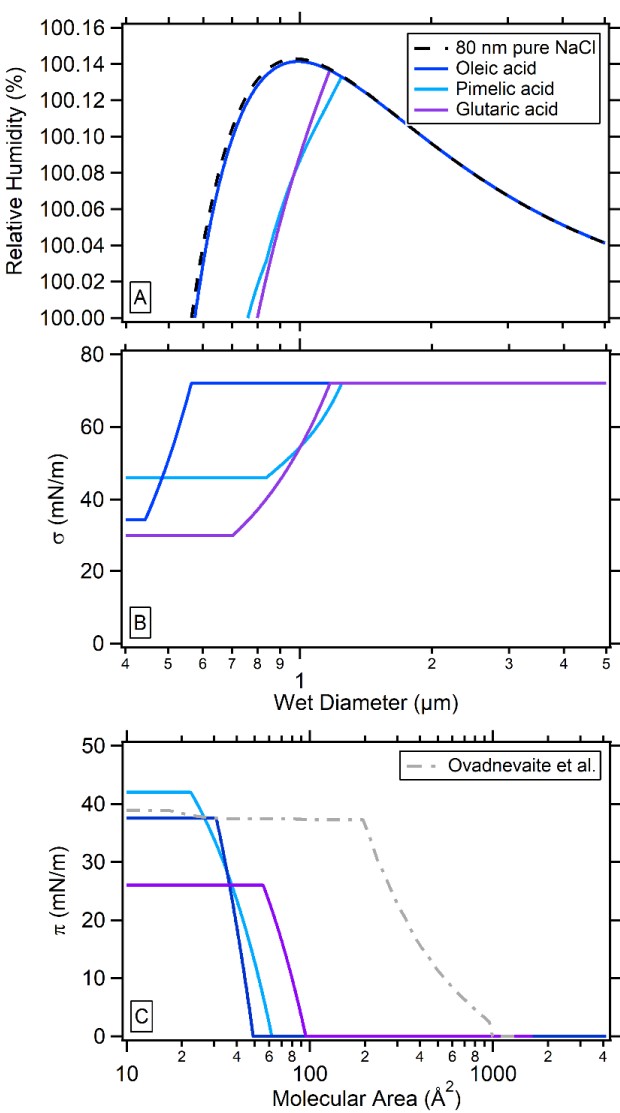

**Figure 7.** Film model derived **(A)** relative humidity and **(B)** surface tension ($\sigma$) as a function of wet diameter for uncoated 80 nm NaCl particles (dashed black line) or 80 nm NaCl particles coated with oleic acid (dark blue line), pimelic acid (light blue line), and glutaric acid (purple line) until $\varepsilon_{org} = 0.80$. Film model parameters for oleic acid are from this study, while the parameters for glutaric acid and pimelic acid were taken from Ruehl et al. (2016). **(C)** Surface pressure ($\pi$) as a function of molecular area for the coated-particle model systems in (A) and (B). Also shown is the $\pi$ curve derived from in Ovadnevaite et al. (2017) based on the reported wet diameter and $\sigma$, assuming $\rho_{org} = 1.6$ g cm$^{-3}$ and a molecular weight = 332 g mol$^{-1}$.



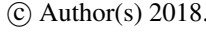

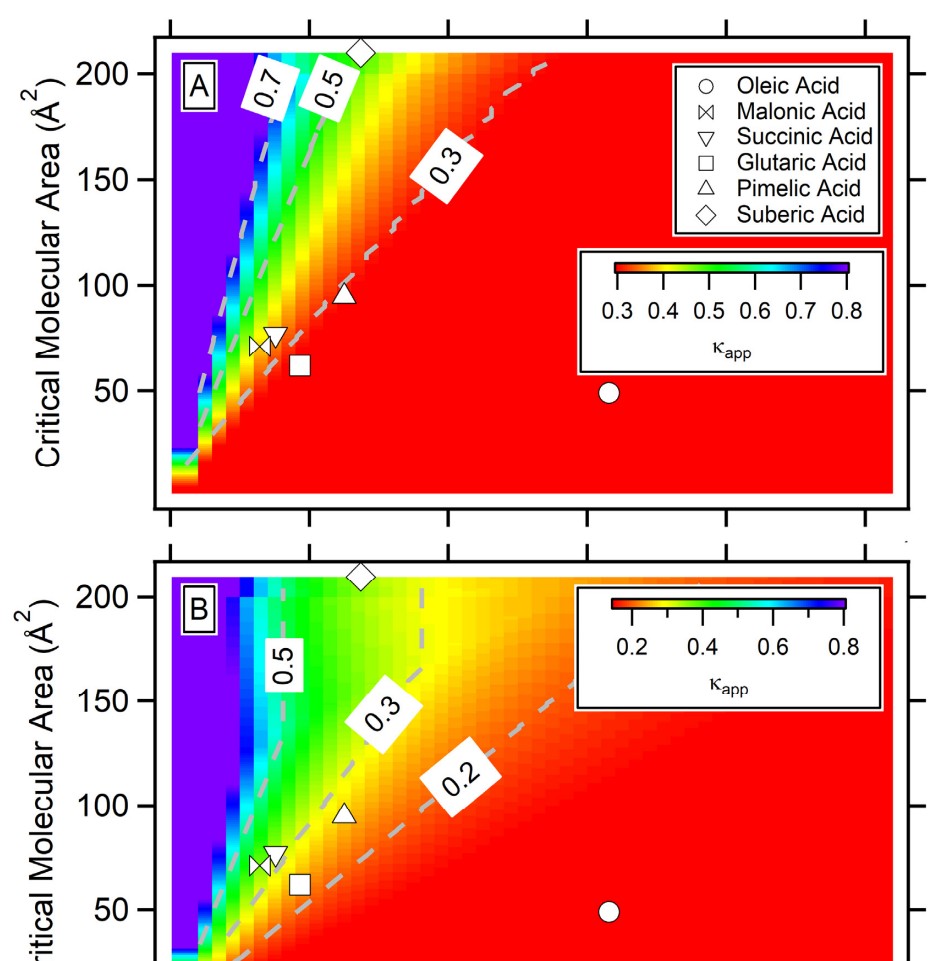

**Figure 8.** Apparent $\kappa$ values calculated as function of critical molecular area (Å$^2$) and molar volume (m$^3$ mol$^{-1}$) from the film model, with the color corresponding to $\kappa_{app}$. Results are shown for 80 nm NaCl particles coated to organic fractions ($\varepsilon_{org}$) of either **(A)** 0.80 and **(B)** 0.90. Also shown are the observed film model fit parameters for oleic acid (circle) from this study. Parameters for glutaric acid (square), succinic acid (upside down triangle), glutaric acid (square), pimelic acid (triangle), and suberic acid (diamond) are from Ruehl et al. (2016). The color scales for panels A and B range from 0.25 to 0.8 and 0.15 to 0.8, respectively, with the color scale saturating at 0.8.



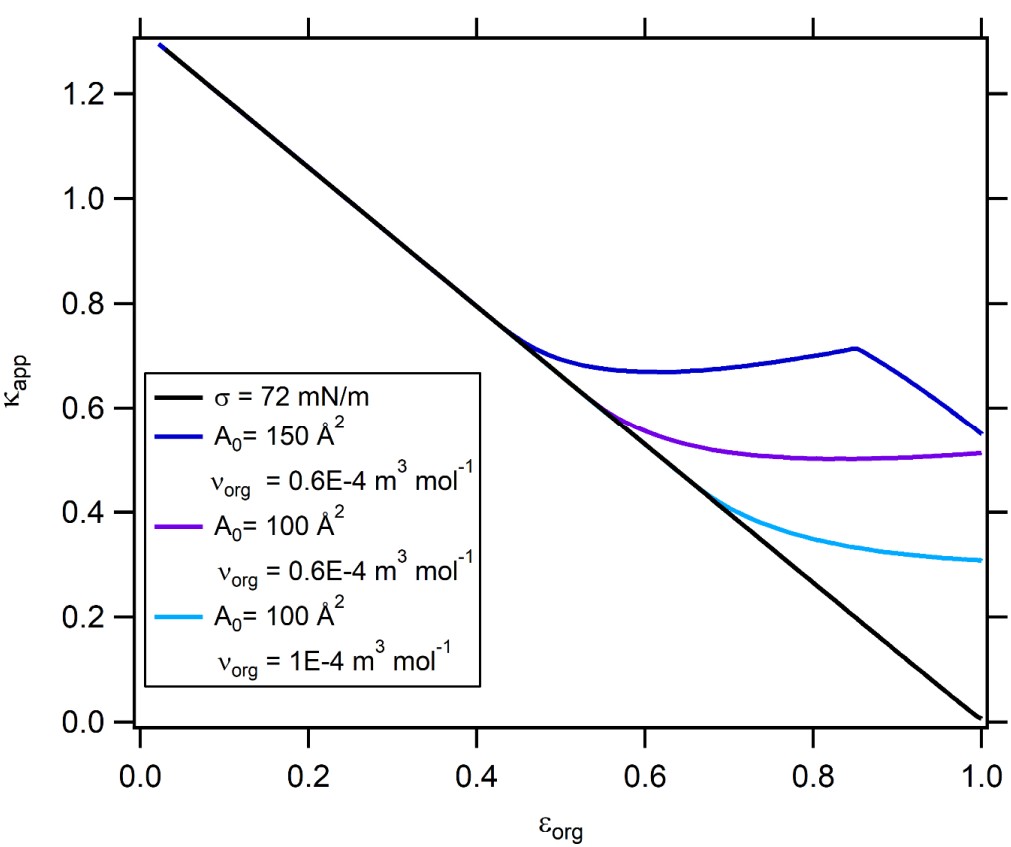

**Figure 9.** Apparent $\kappa$ ($\kappa_{app}$) as a function of $\varepsilon_{org}$ for mixed NaCl-organic particles having a constant total diameter (136.8 nm), calculated using the film model. The black line is the constant $\sigma$ case, the dark blue line was calculated using $A_0 = 150$ Å$^2$ and $\nu_{org} = 0.6 \times 10^{-4}$, the purple line is calculated using $A_0 = 100$ Å$^2$ and $\nu_{org} = 0.6 \times 10^{-4}$, and the light blue line is calculated using $A_0 = 100$ Å$^2$ and $\nu_{org} = 1 \times 10^{-4}$.