# Peer review of "Establishing the Impact of Model Surfactants on Cloud Condensation Nuclei Activity of Sea Spray Aerosol Mimics"

_Atmospheric Chemistry and Physics, 2018_

## Author Comment (AC1) · 17 Mar 2018

It has come to our attention that the particle diameter used in the calculations and experiments in Fig. 5b was incorrectly reported as 180 nm in the caption. The actual seed particle mobility diameter used was 150 nm, which after accounting for the NaCl shape factor, is equivalent to a diameter for use in the Kohler calculations of 142 nm. A corrected caption is provided below.

We thank Prof. Andreas Zuend (McGill Univ.) for pointing out the discrepancy between the figure and the information in the caption.

Figure 5. (A) Apparent $\kappa$ values as a function of $\varepsilon$org as calculated from the critical supersaturation for NaCl particles coated with oleic acid (circles), a mixture of myristic acid and oleic acid (squares), a mixture of palmitic acid and oleic acid (crosses) and oxidized oleic acid (triangles). Also shown is the predicted $\kappa$ based on volume mixing rules assuming that $\kappa$NaCl = 1.3 and $\kappa$org = 0.001, with $\sigma$ = 72 mN/m. Each point is colored by the actual surface tension required to match observations in $\kappa$. (B) Relative humidity as a function of measured wet diameter for 150 nm (mobility diameter) NaCl seed particles coated with a fixed amount of oleic acid ($\varepsilon$org = 0.95). Points are colored by upper-limit estimates for surface tension and the size of the points corresponds to the wet diameter. Also shown are values predicted from the compressed film model and $\kappa$-kohler theory assuming $\sigma$ = 72 mN/m.

---

## Referee Comment (RC1) · Anonymous Referee #1 · 3 Apr 2018

The authors present a comprehensive study on the potential role of a subset of surfactant molecules in dictating aerosol activation behavior. This is an issue that waxes and wanes in the literature, with no consistent theoretical framework or set of standards for comparisons. I appreciate the brief summary of studies presented and I enjoyed digesting the results. In many occasions I found myself deleting a specific point to raise that was answered further in the document. Indeed, the small number of questions raised below reflects this. The authors have considered multiple angles and it is certainly worthy of publication in ACP. I would appreciate that, prior to publication, a number of general issues are clarified. For studies with relatively low number of compounds, theoretical results are more sensitive to variations in pure component and

mixture properties. These issues might have arisen from a confused interpretation on my part, but are important to consider given the varying prescribed importance of this area of study.

Section 2.2 Kappa values are not constant with changing water content due to non-ideal mixing. A value of 1.33 assumes ideality, presumably, whilst Kappa values for Sodium Chloride, theoretically, stride from ~1.5 to 1.23 as RH increases from 90 to 99%RH. Does this affect the methodology for RH calibration? Would this change figure 5B

Equation 2, as used from the Petters and Kreidenweis 2007 paper, is referenced as being valid in that original source when Kappa > 0.2. However, in this study you pre-sented derived Kappa values < 0.02 even for the oxidised test case. I would like to see presentation of why this assumption is not important.

More generally, I'm unsure if the role of activity coefficients in dilute solutions might somehow affect any re-partitioning, assumptions of ideal mixing in equation 4, or raise the potential for LLE in the systems studied here. Could the authors comment?

Section 2.5. Perhaps I've interpreted this section incorrectly, but how are variables used in the fitting process constrained?

Section 3.3 For such high mass fractions, how would any solubility limitations affect results from assuming an ideal kappa-mixing rule?

Section 3.4 Following the previous comments, for the oxidation experiments the authors note that all of the organic mass could be accounted for by three compounds [86+7+7]. All of these compounds still have low kappa values, the highest < 0.02. Multiple studies rely on the potential for LLE to explain any variation in observed hygroscopicity. Given the often high mass fractions of total organic in these studies, I re-iterate if the authors have considered any contribution from a possible LLE, with unknown morphology, that might introduce errors associated with the approach presented here.

Minor comments: Page 11, line 7: Please correct 'The surface tension is constrained by always be larger than some minimum value..'

---

## Author Comment (AC2) · 9 Apr 2018

**Response to Reviewer #1**

We thank the reviewer for the thoughtful comments. We address each comment individually below, with the reviewers initial comment in **black** and our responses in **blue**.

The authors present a comprehensive study on the potential role of a subset of surfactant molecules in dictating aerosol activation behavior. This is an issue that waxes and wanes in the literature, with no consistent theoretical framework or set of standards for comparisons. I appreciate the brief summary of studies presented and I enjoyed digesting the results. In many occasions I found myself deleting a specific point to raise that was answered further in the document. Indeed, the small number of questions raised below reflects this. The authors have considered multiple angles and it is certainly worthy of publication in ACP. I would appreciate that, prior to publication, a number of general issues are clarified. For studies with relatively low number of compounds, theoretical results are more sensitive to variations in pure component and mixture properties. These issues might have arisen from a confused interpretation on my part, but are important to consider given the varying prescribed importance of this area of study.

Section 2.2 Kappa values are not constant with changing water content due to nonideal mixing. A value of 1.33 assumes ideality, presumably, whilst Kappa values for Sodium Chloride, theoretically, stride from ~1.5 to 1.23 as RH increases from 90 to 99%RH. Does this affect the methodology for RH calibration? Would this change figure 5B.

The reviewer raises an important question about the value of kappa that is used here. Indeed, kappa is not fundamentally a constant but varies with RH due to changes in activity. The numbers given by the reviewer are consistent with those calculated from the UManSysProp (http://umansysprop.seaes.manchester.ac.uk/) online tool, using the "Hygroscopic Growth Factors" (HGF) tool. When the "CCN Activation Potential" tool is used the kappa values at the point of activation can be calculated. These calculations indicate that above 99% RH (the upper limit in the "HGF" tool) the kappa values for NaCl again increase and depend on the size of the dry particle, increasing with particle size. For the vast majority of experiments in our study, including calibration experiments, the dry particle size was 180 nm. The UManSysProp calculations indicate that kappa = 1.295 for 150 nm and 1.30 for 180 nm dry particles at the point of activation, RH = 100.04%. This is identical to the value of 1.30 assumed by us. The calibrations occur at > 99% RH for the CFSTGC, typically 99.9% or so (see Figure S1). And the calibrations for the CCN instrument are at >100% RH. Thus, the value of 1.3 is generally reasonable for the conditions here. Further, we can assess the impact of kappa being somewhat smaller, specifically equal to the value at 99% RH (= 1.23). The critical supersaturation would change from 0.0624% (using kappa = 1.3) to 0.0642% (using kappa = 1.23). This is illustrated in the figure below. The difference is exceptionally small, and would not impact our general or specific conclusions. In our revised manuscript, we intend to make note of this issue.

[Figure]

Equation 2, as used from the Petters and Kreidenweis 2007 paper, is referenced as being valid in that original source when Kappa > 0.2. However, in this study you presented derived Kappa values < 0.02 even for the oxidised test case. I would like to see presentation of why this assumption is not important.

To test this assumption we have recalculated the kappa values using the full expression, where the kappa is derived from the critical supersaturation using Eqn. 6 in Petters and Kreidenweis (2007), i.e. from the full Kohler curve. We find that the difference between the kappa from the full expression differs negligibly from that using the approximation (Eqn. 2), with a maximum difference of <0.5% (corresponding to $\Delta\kappa$ ~ 0.0004) of the actual kappa (see figure below). Thus, we see that this assumption is not important, at least for the experiments here. In our revised manuscript we intend to make note of this issue.

[Figure]

More generally, I'm unsure if the role of activity coefficients in dilute solutions might somehow affect any re-partitioning, assumptions of ideal mixing in equation 4, or raise the potential for LLE in the systems studied here. Could the authors comment?

And

Section 3.4 Following the previous comments, for the oxidation experiments the authors note that all of the organic mass could be accounted for by three compounds [86+7+7]. All of these compounds still have low kappa values, the highest < 0.02. Multiple studies rely on the potential for LLE to explain any variation in observed hygroscopicity. Given the often high mass fractions of total organic in these studies, I re-iterate if the authors have considered any contribution from a possible LLE, with unknown morphology, that might introduce errors associated with the approach presented here.

LLE (liquid-liquid equilibrium) or equivalently liquid-liquid phase separation (LLPS). Quoting Ovadnevaite *et al.* (2017), "Liquid-liquid phase separation emerges as a consequence of a substantial degree of non-ideal mixing to the point where the formation of an additional phase becomes thermodynamically favourable (stable liquid-liquid equilibrium state) in comparison to a single, homogenous liquid phase." Very likely LLPS is playing a role for the systems investigated here, with an aqueous-rich phase and an organic-rich phase occurring. Various studies have established that LLPS readily occurs for organic-salt systems when the O:C of the organic compound or mixture is less than ~0.6 (e.g. Bertram *et al.*, 2011; Song *et al.*, 2012). The O:C of the fatty acids (prior to oxidation) used in our study are <0.15. Thus, we fully expect LLPS to occur.

To a reasonable extent, our assumption in the "upper limit" surface tension case is effectively a statement of there being two phases, i.e. of LLPS: an inorganic-aqueous core and an organic shell, the thickness of which depends on the amount of organic and the size of the droplet. This is, in essence, equal to the "model III" in Ovadnevaite et al. (2017). As they describe it, in their simplified organic film model "all organic material resides in a water-free insoluble organic film adsorbed to an aqueous, salt-rich core phase, coating the core phase completely up to the point where a minimum thickness $\delta_{org}$ ($\delta_{\beta}$, min in our model) is reached. Hence, this simplified model resembles our more complex thermodynamic equilibrium model with LLPS, but with two major differences: (1) no organic material dissolves into the growing aqueous phase and (2) no water partitions to the organic film 'phase', that is, a permanent and complete organic–inorganic LLPS at all relative humidity levels." The compressed film model is ultimately just a variation on this same idea, but that allows for some dissolution of the organic in the aqueous, salt-rich core phase at high coverage and that specifies a relationship between the achievable surface tension and bulk concentration (Eqn's 7 & 8 in our paper).

So, in essence, we have considered LLPS. We have not explicitly considered the issue of "unknown morphology," but to the extent that morphology is even really reflected in our model it is in the parameters that are derived. We calculate, for example, molecular area assuming that the organic phase is equally present across the entire droplet surface. We could have alternatively assumed that the organic phase covered e.g. only half of the droplet, making a hemisphere (illustrated in the cartoon below, taken from Ovadnevaite et al. (2017)). Or "molecular islands" could form as the droplet grows and the surface coverage decreases, leaving organic-rich patches where the surface tension remains reduced even if the overall coverage, calculated based on the total surface area of the droplet, is low. Had we done this the molecular area values would change and we would have to calculate the effective surface tension of the droplet as some weighted average of the surfaces exposed to air (i.e. the organicair interface and the water-air interface). This would result in shifts in the molecular area term, and in particular the critical molecular area. And if we calculated the surface tension of just the organic-air interface we would obtain values that would deviate from that obtained assuming full surface coverage, with the "hemisphere" values being smaller for a given total coverage (to maintain the overall average surface tension). Put another way, we fully expect that a more thorough consideration of impacts of morphology (e.g. "islands") would lead us to the same general conclusions, but the derived molecular parameters might change. That said, given that for the oleic acid + salt system we find that the derived surface pressure isotherms are quite similar to bulk experiments, there is no strong reason to think that, at least for the single organic component systems considered here, that the introduction of a more complex morphology model, with unknown parameters, would be beneficial.

In our revised manuscript, we will briefly discuss the relationship between the models that we have used here and the concept of liquid-liquid phase separation.

[Figure]

Section 2.5. Perhaps I've interpreted this section incorrectly, but how are variables used in the fitting process constrained?

Only the variables $A_0$, $m_\sigma$, and $C_0$ are allowed to vary during the fitting process. (Other variables are inputs from the observations). The individual parameters are not explicitly constrained beyond reasonable physical constraints. In particular, all three parameters are constrained to be > 0. However, in a practical sense we have not found it necessary to actually impose these constraints during the data fitting since the natural solution has these parameters in their physically realistic ranges. We do, however, constrain the surface tension to be >0 mN/m. We will make note in the revised manuscript that the parameters are constrained to be within physically reasonable ranges.

Section 3.3 For such high mass fractions, how would any solubility limitations affect results from assuming an ideal kappa-mixing rule?

We may be misinterpreting this question, but we explicitly assume that the organic material is effectively insoluble in water (kappa = 0.001) and thus there are not really "solubility limitations." In the ideal mixing case, the difference between kappa = 0 and kappa = 0.001 is negligible and unnoticeable on a figure such as Fig. 5a. However, to the extent that the organic material is actually somewhat more soluble than we assumed at the high mass fractions, we would expect the calculated kappa from ideal mixing to be too low. In other words, the line shown in figure 5a can be taken as a lower limit.

Minor comments: Page 11, line 7: Please correct 'The surface tension is constrained by always be larger than some minimum value…'

Done.

**References:**

Bertram, A. K., Martin, S. T., Hanna, S. J., Smith, M. L., Bodsworth, A., Chen, Q., et al. (2011), Predicting the relative humidities of liquid-liquid phase separation, efflorescence, and deliquescence of mixed particles of ammonium sulfate, organic material, and water using the organic-to-sulfate mass ratio of the particle and the oxygen-to-carbon elemental ratio of the organic component, *Atmos. Chem. Phys.*, *11*(21), 10995-11006. https://doi.org/10.5194/acp-11-10995-2011

Ovadnevaite, J., Zuend, A., Laaksonen, A., Sanchez, K. J., Roberts, G., Ceburnis, D., et al. (2017), Surface tension prevails over solute effect in organic-influenced cloud droplet activation, *Nature*, *546*(7660), 637-641. https://doi.org/10.1038/nature22806

Song, M., Marcolli, C., Krieger, U. K., Zuend, A., & Peter, T. (2012), Liquid-liquid phase separation in aerosol particles: Dependence on O:C, organic functionalities, and compositional complexity, *Geophysical Research Letters*, *39*(19). https://doi.org/10.1029/2012GL052807

---

## Referee Comment (RC2) · Anonymous Referee #2 · 8 May 2018

Forestieri present a study of droplet growth and CCN activation for NaCl particles that are coated with surface active, hydrophobic organic acids. The data are fit to a surface film model that accounts for surface tension depression of the drop. Comparisons of surface from the film model to bulk material are made. The main experimental finding of the paper is that surface tension is reduced, but measured critical superaturations are minimally affected.

This is a timely and excellent paper. To date very few studies have reported equilibrium growth curves all the way to the point of CCN activation. Furthermore, to the paper shows direct comparisons to bulk surface pressure measurements, a link that

has been difficult to establish. The paper is well written and well referenced. I recommend publication as is.

I would like to add one comment. The calculations in Figure 9 show that the surface film model predicts that pure hydrophobic organic compounds could have an apparent kappa as large as 0.5. The manuscript does make clear that this has not been observed in their data and that this may explain the data of Collins et al. (2016). The authors rightfully point out these authors did not measure composition, making it difficult to perform a direct comparison to theory. Overall, I think the discussion in the manuscript on this point is well balanced.

There have been various claims made about the role of surfactants in droplet activation for some number of years now, with some suggesting that surface tension can cause pure or nearly pure organic compounds to have kappa values that rival those of inorganics, as also suggested by Figure 9. This presents a simple question. Are there any direct measurements (where one is confident about particle composition) that would obey any the three surfactant model lines shown? If the answer is no, how would one guide modelers to constrain the film theory to prevent such unphysical predictions? Or phrased differently, how would the authors review a hypothetical future study that couples the film model with a cloud droplet parameterization, tunes A0, and determines the sensitivity of aerosol indirect effects to the presence of surface active organics?

---

## Referee Comment (RC3) · Anonymous Referee #3 · 14 May 2018

This manuscript presents a high quality experimental and modeling study of the effect of coatings of organic surfactants on monodisperse NaCl particles. This is a relevant and timely piece of work adding interesting insight to the discussion of the role of surfactants in cloud droplet activation and explaining literature observations that highly surface active fatty acids have little effect on CCN activity.

I enjoyed reading this manuscript and recommend publication after minor revision.

Comments and suggestions

Abstract: in the abstract it is stated that a kinetic limitation to water uptake is observed for NaCl particles coated with pure palmitic acid, I suggest the coating thickness is

provided.

Introduction

Page 4: There are several studies relevant on NaCl, surfactants and ccn activity showing different effects which could be included in the discussion – e.g. Petters and Petters J. Geophys. Res. Atmos.121 , 1878, King et al. Environ. Sci. Technol. 2012, 46, 10405. The different results obtained illustrate the importance of a better understanding of surfactants.

Methods Page 5: line 23. It should be specified (given in nm) what is meant by "thin" and "thick" coatings respectively.

Description of coating experiments: I suggest to include a schematic of the flow tube and associated dilution system in Figure 1.

How can the residence time of 0 s be understood? That would be injection and sampling at the same spot? Why is that discussed?

Perhaps I misunderstand something, but if the concentration of ozone is 278 ppb in the flow tube and the pressure is atmospheric pressure ($\sim 2.46 \times 10^{19}$ molecules per cc) how can the ozone exposure be so high as $10^{14}$ molecules per cc? ($278 \times 10^{-9} \times 2.46 \times 10^{19} \sim 7 \times 10^{12}$ molecules per cc) – is there a dilution effect involved?

Regarding calibration: reviewer 1 has already raised the subject that kappa values are not constant with particle size and I agree that this should be addressed in the revised manuscript. In their reply the authors have shown that this is not relevant for their particles sizes for NaCl. What about ammonium sulfate? On a related note shape factors of 1.08 and 1.04 are included in the calibration of the CFSTGC – is it based on the literature or have shape factors for the specific system used by the authors been measured?

Was a shape factor also applied in the calibration of the CCN counter? This should be stated.

It would be nice to see a figure with the particle size distributions before and after coating – in some cases the coatings are very thick - how broad were the size distributions after coating?

Some references for the equations given and the underlying assumptions would be nice, e.g. equation (6) is it assumed that all organics are at the surface? Relation between equation 6 and 10 –somewhat overlapping but have different symbols (A and A_mic). I don't think the symbol for molecular volume of the organic is explained in relation to equation 10.

It is a bit difficult to get an overview of the experiments actually performed. In Table 1 it just says the type of coating. How many different dry sizes, RH values and coating thicknesses form the basis for each set of fitted parameters?

As I understand f_surf was an assumed parameter - I miss a discussion on the importance of f_surf and what values of f_surf were assumed in relation to the parameters Table 1.

Perhaps I have overlooked it, but I could not find quantitative information on the magnitude of the uncertainly on RH – it says "Uncertainty in the film model curve was estimated by perturbing the input RH values by the average precision-based uncertainty in RH." – what was the actual average precision-based uncertainly in RH?

Linking to sea spray aerosol and secondary marine aerosol: I find the discussion a bit hard to follow – what is meant when referring to "high" CCN activation efficiency, it says kappa_app >0.7 – but 0.7 is lower than the kappa for artificial sea salt which is around 0.9? Some literature references should be given to support the statement "The often large CCN activation efficiency for particles observed in the marine environment, in particular nascent SSA particles". Perhaps also references to work showing other trends should be included; for comparison, it has been reported that particles generated from sea water or sea water spiked with sea surface microlayer had CCN activity similar to sea salt (Rasmussen et al. 2017, J. Aerosol Sci. 107, 134) and high

kappa values in ambient environment can also be related to organic-to-sulfate ratio (Yakobi-Hancock et al. 2014 , Atmos. Chem. Phys., 14, 12307). Since also aged and secondary particles are addressed, it could also be mentioned that surfactants may come from the gas phase (Sareen et al. Proc Natl Acad Sci U S A. 2013, 110, 272).

Minor Title: The title says "Sea spray aerosols" – I suggest changing to "model sea spray", "salts" or "Sea spray aerosol mimics" or similar since the study was performed on NaCl core particles.

In the caption for Table 1 caption it says: "Uncertainties on the film model parameters on the precision in RH (see text for details)" – it is not clear to me what is meant here.

Figure S2: it should say in the figure caption that the calibration was performed with NaCl particles.

Figure 2: figure text- "confidence interval" – is this 95% ? should be stated.

---

## Author Comment (AC3) · 13 Jun 2018

We thank the reviewers for the thoughtful comments. We address each comment individually below, with the reviewers initial comment in **black** and our responses in **blue**.

**Response to Reviewer #1**

We have slightly expanded our response to Reviewer #1, relative to our original posting, to illustrate the specific changes made to the manuscript in response to the comments.

The authors present a comprehensive study on the potential role of a subset of surfactant molecules in dictating aerosol activation behavior. This is an issue that waxes and wanes in the literature, with no consistent theoretical framework or set of standards for comparisons. I appreciate the brief summary of studies presented and I enjoyed digesting the results. In many occasions I found myself deleting a specific point to raise that was answered further in the document. Indeed, the small number of questions raised below reflects this. The authors have considered multiple angles and it is certainly worthy of publication in ACP. I would appreciate that, prior to publication, a number of general issues are clarified. For studies with relatively low number of compounds, theoretical results are more sensitive to variations in pure component and mixture properties. These issues might have arisen from a confused interpretation on my part, but are important to consider given the varying prescribed importance of this area of study.

Section 2.2 Kappa values are not constant with changing water content due to nonideal mixing. A value of 1.33 assumes ideality, presumably, whilst Kappa values for Sodium Chloride, theoretically, stride from ~1.5 to 1.23 as RH increases from 90 to 99%RH. Does this affect the methodology for RH calibration? Would this change figure 5B.

The reviewer raises an important question about the value of kappa that is used here. Indeed, kappa is not fundamentally a constant but varies with RH due to changes in activity. The numbers given by the reviewer are consistent with those calculated from the UManSysProp (http://umansysprop.seaes.manchester.ac.uk/) online tool, using the "Hygroscopic Growth Factors" (HGF) tool. When the "CCN Activation Potential" tool is used the kappa values at the point of activation can be calculated. These calculations indicate that above 99% RH (the upper limit in the "HGF" tool) the kappa values for NaCl again increase and depend on the size of the dry particle, increasing with particle size. For the vast majority of experiments in our study, including calibration experiments, the dry particle size was 180 nm. The UManSysProp calculations indicate that kappa = 1.295 for 150 nm and 1.30 for 180 nm dry particles at the point of activation, RH = 100.04%. This is identical to the value of 1.30 assumed by us. The calibrations occur at > 99% RH for the CFSTGC, typically 99.9% or so (see Figure S1). And the calibrations for the CCN instrument are at >100% RH. Thus, the value of 1.3 is generally reasonable for the conditions here. Further, we can assess the impact of kappa being somewhat smaller, specifically equal to the value at 99% RH (= 1.23). The critical supersaturation would change from 0.0624% (using kappa = 1.3) to 0.0642% (using kappa = 1.23). This is illustrated in the figure below. The difference is exceptionally small, and would not impact our general or specific conclusions. Similar arguments can be made for ammonium sulfate. There is a slightly larger impact of uncertainty in kappa at lower RH. The greater sensitivity is related to the steeper slope in the RH versus wet diameter relationship at RH values lower than the critical supersaturation. Assuming an uncertainty of ±0.05 in κ corresponds to an uncertainty in RH of ca. 0.01% (for NaCl) and 0.02% (for ammonium sulfate) around 99.9% RH.

We have updated the text in section 2.2 to include brief discussion of these issues.

> *"The RH values were calculated from Eqn. 1 assuming that κ values were 1.3 and 0.61 for NaCl and ammonium sulfate, respectively (Petters and Kreidenweis, 2007). κ values can vary with particle size and RH, due to variations in activity. The use of these specific values is reasonable for the particle sizes (>100 nm) and RH range (>99.9% RH) considered in our experiments (Topping et al., 2016). An uncertainty of ±0.05 in κ corresponds to an uncertainty in RH of ca. 0.01% (for NaCl) and 0.02% (for ammonium sulfate) around 99.9% RH, but of <0.01% at RH values near the critical supersaturation."*

[Figure]

Equation 2, as used from the Petters and Kreidenweis 2007 paper, is referenced as being valid in that original source when Kappa > 0.2. However, in this study you presented derived Kappa values < 0.02 even for the oxidised test case. I would like to see presentation of why this assumption is not important.

To test this assumption we have recalculated the kappa values using the full expression, where the kappa is derived from the critical supersaturation using Eqn. 6 in Petters and Kreidenweis (2007), i.e. from the full Kohler curve. We find that the difference between the kappa from the full expression differs negligibly from that using the approximation (Eqn. 2), with a maximum difference of <0.5% (corresponding to $\Delta\kappa \sim 0.0004$) of the actual kappa (see figure below). Thus, we see that this assumption is not important, at least for the experiments here. We have added the following in Section 2.2:

> *"Petters and Kreidenweis (2007) note that Eqn. 2 is only valid when k > 0.2. The use of Eqn. 2 will not impact our calibrations, given the use of NaCl. The use of Eqn. 2 for determination of $\kappa_{app}$ for experiments with particles having large organic fractions, for which $\kappa_{app} < 0.2$, can lead to a small overestimate in $\kappa_{app}$. Comparison of the $\kappa_{app}$ determined from Eqn. 2 compared to from the full expression indicates that the error introduced is <0.4% in $\kappa_{app}$, corresponding to an absolute*

[Figure]

*value of <0.0003 for $\kappa_{app}$ < 0.07 (approximately the lowest value from the NaCl-organic mixtures used here)."*

More generally, I'm unsure if the role of activity coefficients in dilute solutions might somehow affect any re-partitioning, assumptions of ideal mixing in equation 4, or raise the potential for LLE in the systems studied here. Could the authors comment?

And

Section 3.4 Following the previous comments, for the oxidation experiments the authors note that all of the organic mass could be accounted for by three compounds [86+7+7]. All of these compounds still have low kappa values, the highest < 0.02. Multiple studies rely on the potential for LLE to explain any variation in observed hygroscopicity. Given the often high mass fractions of total organic in these studies, I re-iterate if the authors have considered any contribution from a possible LLE, with unknown morphology, that might introduce errors associated with the approach presented here.

LLE (liquid-liquid equilibrium) or equivalently liquid-liquid phase separation (LLPS). Quoting Ovadnevaite et al. (2017), "Liquid-liquid phase separation emerges as a consequence of a substantial degree of non-ideal mixing to the point where the formation of an additional phase becomes thermodynamically favourable (stable liquid-liquid equilibrium state) in comparison to a single, homogenous liquid phase." Very likely LLPS is playing a role for the systems investigated here, with an aqueous-rich phase and an organic-rich phase occurring. Various studies have established that LLPS readily occurs for organic-salt systems when the O:C of the organic compound or mixture is less than ~0.6 (e.g. Bertram et al., 2011;Song et al., 2012). The O:C of the fatty acids (prior to oxidation) used in our study are <0.15. Thus, we fully expect LLPS to occur.

To a reasonable extent, our assumption in the "upper limit" surface tension case is effectively a statement of there being two phases, i.e. of LLPS: an inorganic-aqueous core and an organic shell, the thickness of which depends on the amount of organic and the size of the droplet. This is, in essence, equal to the "model III" in Ovadnevaite et al. (2017). As they describe it, in their simplified organic film model "all organic material resides in a water-free insoluble organic film adsorbed to an aqueous, saltrich core phase, coating the core phase completely up to the point where a minimum thickness δorg (δβ, min in our model) is reached. Hence, this simplified model resembles our more complex thermodynamic equilibrium model with LLPS, but with two major differences: (1) no organic material dissolves into the growing aqueous phase and (2) no water partitions to the organic film 'phase', that is, a permanent and complete organic–inorganic LLPS at all relative humidity levels." The compressed film model is ultimately just a variation on this same idea, but that allows for some dissolution of the organic in the aqueous, salt-rich core phase at high coverage and that specifies a relationship between the achievable surface tension and bulk concentration (Eqn's 7 & 8 in our paper).

So, in essence, we have considered LLPS. We have not explicitly considered the issue of "unknown morphology," but to the extent that morphology is even really reflected in our model it is in the parameters that are derived. We calculate, for example, molecular area assuming that the organic phase is equally present across the entire droplet surface. We could have alternatively assumed that the organic phase covered e.g. only half of the droplet, making a hemisphere (illustrated in the cartoon below, taken from Ovadnevaite et al. (2017)). Or "molecular islands" could form as the droplet grows and the surface coverage decreases, leaving organic-rich patches where the surface tension remains reduced even if the overall coverage, calculated based on the total surface area of the droplet, is low. Had we done this the molecular area values would change and we would have to calculate the effective surface tension of the droplet as some weighted average of the surfaces exposed to air (i.e. the organic-air interface and the water-air interface). This would result in shifts in the molecular area term, and in particular the critical molecular area. And if we calculated the surface tension of just the organic-air interface we would obtain values that would deviate from that obtained assuming full surface coverage, with the "hemisphere" values being smaller for a given total coverage (to maintain the overall average surface tension). Put another way, we fully expect that a more thorough consideration of impacts of morphology (e.g. "islands") would lead us to the same general conclusions, but the derived molecular parameters might change. That said, given that for the oleic acid + salt system we find that the derived surface pressure isotherms are quite similar to bulk experiments, there is no strong reason to think that, at least for the single organic component systems considered here, that the introduction of a more complex morphology model, with unknown parameters, would be beneficial.

[Figure]

We have added the following in Section 2.4:

"The use of Eqn. 1 provides a lower limit estimate of σ, since it is assumed that none of the organic component partitions into the bulk droplet and is present only at the surface. These values are therefore referred to as the lower-limit σ. *This model essentially assumes a phase-separation between a water-free organic phase and water-rich inorganic phase, mimicking a permanent liquid-liquid phase separation (Ovadnevaite et al., 2017).*"

And in Section 2.5:

"The organic compounds at the air-water interface can contribute to σ depression, while organic compounds dissolved into the bulk contribute to droplet growth through the Roault effect. *Like*

*the upper-limit surface pressure model above, the compressed film model essentially assumes a phase separation between an organic-rich surface layer and a bulk solution, but with the bulk solution now being a ternary water-inorganic-organic mixture. Also, for the compressed film model the distribution of the organic material between the surface layer and the bulk varies with RH. This general behavior is similar to the thermodynamic model of liquid-liquid phase separation considered by Ovadnevaite et al. (2017), although the two models differ in terms of their details.* The film model is a 2-dimensional (2D) equation of state…"

Section 2.5. Perhaps I've interpreted this section incorrectly, but how are variables used in the fitting process constrained?

Only the variables $A_0$, $m_\sigma$, and $C_0$ are allowed to vary during the fitting process. (Other variables are inputs from the observations). The individual parameters are not explicitly constrained beyond reasonable physical constraints. In particular, all three parameters are constrained to be > 0. However, in a practical sense we have not found it necessary to actually impose these constraints during the data fitting since the natural solution has these parameters in their physically realistic ranges. We do, however, constrain the surface tension to be >0 mN/m. We now state:

"The variation of σ with $A$ is solved for from the observations by minimizing the chi-square value for $D_{wet}$ by varying $A_0$, $m_\sigma$, and $C_0$ with $D_{coat}$, $D_{NaCl}$, and $RH$ as inputs. *The surface tension is constrained to always be larger than some minimum value ($\sigma_{min}$ or $\pi_{max}$) that is determined as part of the data fitting; this constraint places bounds on the acceptable values of $A_0$ and $m_\sigma$. The parameters $A_0$, $m_\sigma$, and $C_0$ are also constrained to be physically realistic, that is with $A_0 > 0$ and $m_s$ and $C_0 \geq 0$.* When the system reaches…"

Section 3.3 For such high mass fractions, how would any solubility limitations affect results from assuming an ideal kappa-mixing rule?

We may be misinterpreting this question, but we explicitly assume that the organic material is effectively insoluble in water (kappa = 0.001) and thus there are not really "solubility limitations." In the ideal mixing case, the difference between kappa = 0 and kappa = 0.001 is negligible and unnoticeable on a figure such as Fig. 5a. However, to the extent that the organic material is actually somewhat more soluble than we assumed at the high mass fractions, we would expect the calculated kappa from ideal mixing to be too low. In other words, the line shown in figure 5a can be taken as a lower limit.

Minor comments: Page 11, line 7: Please correct 'The surface tension is constrained by always be larger than some minimum value…'

Done.

**Response to Reviewer #2**

Forestieri present a study of droplet growth and CCN activation for NaCl particles that are coated with surface active, hydrophobic organic acids. The data are fit to a surface film model that accounts for surface tension depression of the drop. Comparisons of surface from the film model to bulk material are

made. The main experimental finding of the paper is that surface tension is reduced, but measured critical superaturations are minimally affected.

This is a timely and excellent paper. To date very few studies have reported equilibrium growth curves all the way to the point of CCN activation. Furthermore, to the paper shows direct comparisons to bulk surface pressure measurements, a link that has been difficult to establish. The paper is well written and well referenced. I recommend publication as is.

We thank the reviewer for the very kind words.

I would like to add one comment. The calculations in Figure 9 show that the surface film model predicts that pure hydrophobic organic compounds could have an apparent kappa as large as 0.5. The manuscript does make clear that this has not been observed in their data and that this may explain the data of Collins et al. (2016). The authors rightfully point out these authors did not measure composition, making it difficult to perform a direct comparison to theory. Overall, I think the discussion in the manuscript on this point is well balanced.

There have been various claims made about the role of surfactants in droplet activation for some number of years now, with some suggesting that surface tension can cause pure or nearly pure organic compounds to have kappa values that rival those of inorganics, as also suggested by Figure 9. This presents a simple question. Are there any direct measurements (where one is confident about particle composition) that would obey any the three surfactant model lines shown? If the answer is no, how would one guide modelers to constrain the film theory to prevent such unphysical predictions? Or phrased differently, how would the authors review a hypothetical future study that couples the film model with a cloud droplet parameterization, tunes A0, and determines the sensitivity of aerosol indirect effects to the presence of surface active organics?

This is something that we have been actively thinking about, but do not yet have a clear answer. Solving for the unknown(s) is simple when the particle composition is known, as it is in these laboratory studies. This becomes much more challenging when dealing with real systems, which introduces both an element of uncertainty (via the unknown) and most likely an greater degree of scatter due to natural variability in the real system. Most likely, greater integration with theory (e.g. the thermodynamic models of Zeund and Dutcher) will be needed, as will continued exploration of multi-component systems of increasing—yet known—complexity. Work such as that by Petters and Petters (2016) helps to point the way as well. The work of Ovadnevaite et al. (2017) also provides some guidance. We think that this will need to be an ongoing conversation across the community.

**Response to Reviewer #3**

This manuscript presents a high quality experimental and modeling study of the effect of coatings of organic surfactants on monodisperse NaCl particles. This is a relevant and timely piece of work adding interesting insight to the discussion of the role of surfactants in

cloud droplet activation and explaining literature observations that highly surface active fatty acids have little effect on CCN activity. I enjoyed reading this manuscript and recommend publication after minor revision.

We thank the reviewer for the kind words.

Comments and suggestions

Abstract: in the abstract it is stated that a kinetic limitation to water uptake is observed for NaCl particles coated with pure palmitic acid, I suggest the coating thickness is provided.

The coating thicknesses for NaCl coated with palmitic acid ranged from 67 nm to 132 nm. Here, coating thickness is defined as $(D_{coated} - D_{seed})/2$. This has been added to the abstract in the revised manuscript.

Introduction
Page 4: There are several studies relevant on NaCl, surfactants and ccn activity showing different effects which could be included in the discussion – e.g. Petters and Petters J. Geophys. Res. Atmos.121 , 1878, King et al. Environ. Sci. Technol. 2012, 46, 10405. The different results obtained illustrate the importance of a better understanding of surfactants.

We have added brief mention of the Petters and Petters (2016) study and of a study by Prisle et al. (2010), who looked at binary NaCl + ionic surfactant mixtures. We did not add mention of King et al. (2012) simply because they do not compare their observed CCN activities to some theoretical explanation. The edited text follows:

> Since this class of organic species is surface active (Schwier et al., 2012) they have the potential to enhance observed CCN activation efficiency by depressing surface tension, but the overall effect of fatty acid addition on CCN activation efficiency of salt particles has been shown to be minimal (Nguyen et al., 2017). *Petters and Petters (2016) additionally investigated binary salt + non-ionic surfactant mixtures and observed that the CCN activation efficiency was, in some cases, actually suppressed relative to ideal mixing. Similarly, Prisle et al. (2010) observed that the CCN activation efficiency of binary NaCl + ionic surfactant mixtures was reasonably well-described assuming the surfactants to be fully soluble with no impact on surface tension.* In this work we will clarify why such highly surface active species *can* have, *and in the case of fatty acids do have*, little impact CCN activation efficiency.

Methods Page 5: line 23. It should be specified (given in nm) what is meant by "thin" and "thick" coatings respectively.

Thin and thick coatings correspond to thicknesses of ~30 and ~80 nm, respectively. This has been added to the revised manuscript.

Description of coating experiments: I suggest to include a schematic of the flow tube and associated dilution system in Figure 1.

The schematic is now included as part of Figure 1 in the revised manuscript.

How can the residence time of 0 s be understood? That would be injection and sampling at the same spot? Why is that discussed?

The referee is correct. For a residence time of 0 seconds, the injector and sampling port are in the same spot. However, due to the non-ideal flow described on Page 6 lines 15-20, the oleic acid coatings for particles sampled in the 0 second configuration were somewhat oxidized. This is noted on Page 21 lines 25-26.

Perhaps I misunderstand something, but if the concentration of ozone is 278 ppb in the flow tube and the pressure is atmospheric pressure ($\sim 2.46 \times 10^{19}$ molecules per cc) how can the ozone exposure be so high as $10^{14}$ molecules per cc? ($278*10^9*2.46*10^{19} \sim 7 \times 10^{12}$ molecules per cc) – is there a dilution effect involved?

The range of exposures given in the manuscript were from 20 seconds to 57 seconds. For example, a 20 second exposure time corresponds to an exposure of $278 \times 10^{-9}*2.46 \times 10^{19}$ (mlcs/cc) * 20 seconds $\sim 1.4 \times 10^{14}$ mlcs/cc *seconds. The 0 second case would have a theoretical exposure 0 mlcs per cc * seconds. The residence times corresponding to this range of exposures is included in the revised manuscript. We have worked to clarify this on Page 6.

> Residence times calculated based on the flow rate and flow tube volume *were 0 s, 20 s, and 57 s*. However, the air was sampled from the side of the flow tube and so the effective residence time was longer and was never as short as 0 s *and the minimum ozone exposure was actually greater than zero*. There was a bypass channel for the flow tube to monitor particle size and hygroscopicity prior to the flow tube experiments; *these bypass measurements correspond to an actual ozone exposure of zero*. The estimated ozone exposure ranged from $1.4 \times 10^{14}$ (*20 seconds*) to $3.9 \times 10^{14}$ (*57 seconds*) molecules cm$^{-3}$ s for non-zero residence times.

Regarding calibration: reviewer 1 has already raised the subject that kappa values are not constant with particle size and I agree that this should be addressed in the revised manuscript. In their reply the authors have shown that this is not relevant for their particles sizes for NaCl. What about ammonium sulfate?

We have repeated this exercise for ammonium sulfate, and added discussion in our extended response to Reviewer #1. In terms of size dependence, for the size ranges considered here (>100 nm), variability in kappa with size is much less important than variability in kappa with RH. The kappa at 99.0% RH (from UManSysProp) for 180 nm ammonium sulfate particles is 0.50, while at activation (= 100.0593% RH) kappa = 0.66. We used kappa = 0.61 in our calculations, a reasonable intermediate value that should be applicable to RH = 99.9%. Uncertainty in kappa for ammonium sulfate has negligible impact on the RH calibration around the critical supersaturation. If it is assumed that kappa = 0.61 vs. 0.66, the critical supersaturations differ by only 0.002%. At lower RH (e.g. 99.9%), uncertainty in kappa does contribute to a greater extent to uncertainty in the RH calibration. For an uncertainty in kappa of +/- 0.05, for ammonium

sulfate this corresponds to an uncertainty in the RH of ca. 0.02%. We now note this in the manuscript.

> "The RH values were calculated from Eqn. 1 assuming that κ values were 1.3 and 0.61 for NaCl and ammonium sulfate, respectively *(Petters and Kreidenweis, 2007). κ values can vary with particle size and RH, due to variations in activity. The use of these specific values is reasonable for the particle sizes (>100 nm) and RH range (>99.9% RH) considered in our experiments (Topping et al., 2016). An uncertainty of ±0.05 in κ corresponds to an uncertainty in RH of ca. 0.01% (for NaCl) and 0.02% (for ammonium sulfate) around 99.9% RH, but of <0.01% at RH values near the critical supersaturation."*

On a related note shape factors of 1.08 and 1.04 are included in the calibration of the CFSTGC – is it based on the literature or have shape factors for the specific system used by the authors been measured?

The shape factor for NaCl was based on the shape factor calculated for a cube (1.08), and constrained by observations (Mikhailov et al., 2004). The assumed shape factor of 1.04 ammonium sulfate particles was also based on literature values (Zelenyuk et al., 2006).We have added these references to the manuscript.

Was a shape factor also applied in the calibration of the CCN counter? This should be stated

Yes, this shape factor was also applied to the calibrations. This has been clarified in the manuscript.

It would be nice to see a figure with the particle size distributions before and after coating – in some cases the coatings are very thick - how broad were the size distributions after coating?

A figure showing three distributions from one experimental day (shown below) was added to the supporting material. The width of the lognormal distribution ranges from 0.07 for the uncoated particles to 0.10 for thickly coated particles.

[Figure]

Some references for the equations given and the underlying assumptions would be nice, e.g. equation (6) is it assumed that all organics are at the surface? Relation between equation 6 and 10 –somewhat overlapping but have different symbols (A and A_mic). I don't think the symbol for molecular volume of the organic is explained in relation to equation 10.

For Eqn. 6, this is simply derived from geometry and thus no reference is provided. However, we have clarified the assumption inherent (that all organics are at the surface). "*The molecular area from Eqn. 6 is a lower limit, as it assumes that all of the surfactant is at the surface.*"

We have also clarified that the $A_{mlc}$ from Eqn. 6 is not necessarily the same as the $A$ used in Eqn.'s 7-10. In particular, $A \geq A_{mlc}$.

For the other key equations (7-10), these are explained in (Ruehl et al., 2016) in detail. Rather than repeat the discussion in that paper, we have added text to make it clearer that this information is available. "The compressed film model was used for this purpose, *and a full description of the equations and underlying assumptions are available in the Supplementary Material of Ruehl et al. (2016).*" That said, we do note that after both Eqn. 7 and Eqn. 10 we have various sentences that include the phrase (at least approximately), that "…the film model assumes…" We have also added text, per reviewer #1's comments, to describe the relationship between the compressed film and upper-limit surface pressure models and the concept of liquid-liquid phase separation.

It is a bit difficult to get an overview of the experiments actually performed. In Table 1 it just says the type of coating. How many different dry sizes, RH values and coating thicknesses form the basis for each set of fitted parameters?

This information has been added to Table 1 in the revised manuscript.

**Table 1.** Summary of experiments and corresponding compressed film model fit parameters. Uncertainties on the film model fit parameters on the precision in RH (see text for details).

| Coating | Range of $\varepsilon_{org}$ | RH Range | $D_{seed}$ (nm) | Oleic Acid Fraction | $A_0$ (Å$^2$) | $m_\sigma$ (mJ m$^{-2}$) | log $C_0$ (mol mol$^{-1}$) | $\pi_{max}$ (mN/m) |
|---|---|---|---|---|---|---|---|---|
| Oleic Acid (N = 125) | 0.42 - 0.97 | 99.83 – 100.03% | 150, 180, 200 | 1 | 48.5 ± 3.8 | 2.15 ± 0.10 | -6.10 ± 0.38 | 37.5 ± 4.8 |
| Myristic Acid (N = 43) | 0.40 – 0.98 | 99.83 – 99.93% | 180, 200 | 1 | 29.2 ± 1.2 | 1.28 ± 0.20 | -7.40 ± 0.08 | ** |
| Oleic Acid and Palmitic Acid (N = 41) | 0.58 – 0.95 | 99.86 – 100.00% | 180, 200 | 0.33 ± 0.08 | 35.2 ± 3.3 | 5.41 ± 0.88 | -5.24 ± 0.22 | 43.6 ± 6.3 |
| Oxidized Oleic Acid[#] (N = 5) | 0.86 – 0.88 | 99.97 – 99.99% | 200 | % | # | | | |
| Glutaric Acid[&] | | | | | 62 | 1.06 | -7 | 42 |
| Pimelic Acid[&] | | | | | 95 | 0.66 | -6 | 26 |

[#] These experiments were conducted only at one $\varepsilon_{org}$ and thus the data are too limited for application of the film model
** No minimum surface tension was reached for this experiment
[&] From Ruehl et al. (2016)
[%]The oleic acid fractions for this coating were variable

As I understand f_surf was an assumed parameter - I miss a discussion on the importance of f_surf and what values of f_surf were assumed in relation to the parameters Table 1.

*$f_{surf}$ is not an assumed parameter. Rather $A$ and $C_{bulk}$ are functions of $f_{surf}$ (a single unknown variable), which is solved for numerically. This has been clarified in the revised manuscript. We have added: "The parameters $A$ and $C_{bulk}$ are related through a single unknown, $f_{surf}$, which is solved for numerically. The parameter $f_{surf}$ empirically characterizes the fraction of the organic species that reside at the particle surface, with $1-f_{surf}$ corresponding to the fraction that is dissolved in the bulk."*

Perhaps I have overlooked it, but I could not find quantitative information on the magnitude of the uncertainly on RH – it says "Uncertainty in the film model curve was estimated by perturbing the input RH values by the average precision-based uncertainty in RH." – what was the actual average precision-based uncertainly in RH?

The average precision-based uncertainty for different experiments ranged from 0.008% to 0.01%. This is now clarified in Section 2.5. This precision-based estimate is reasonably

consistent with the uncertainty that results from uncertainty in kappa values of the calibration salts, given the good agreement between the NaCl and ammonium sulfate calibrations.

Linking to sea spray aerosol and secondary marine aerosol:  I find the discussion a bit hard to follow – what is meant when referring to "high" CCN activation efficiency, it says kappa_app >0.7 – but 0.7 is lower than the kappa for artificial sea salt which is around 0.9?  Some literature references should be given to support the statement "The often large CCN activation efficiency for particles observed in the marine environment, in particular nascent SSA particles".  Perhaps also references to work showing other trends should be included;  for comparison, it has been reported that particles generated from sea water or sea water spiked with sea surface microlayer had CCN activity similar to sea salt (Rasmussen et al. 2017, J. Aerosol Sci. 107, 134) and high kappa values in ambient environment can also be related to organic-to-sulfate ratio (Yakobi-Hancock et al. 2014 , Atmos. Chem. Phys., 14, 12307). Since also aged and secondary particles are addressed, it could also be mentioned that surfactants may come from the gas phase (Sareen et al. Proc Natl Acad Sci U S A. 2013, 110, 272).

The use of the descriptor "high" is relative. We mean this in the context of much higher than expected given the likely large organic content of the particles. We attempted to provide this context through, e.g. the (now modified) second sentence of the following:

> As another example, Collins et al. (2016) observed persistently high CCN activation efficiency ($\kappa_{app} > 0.7$) for microcosm studies of nascent SSA with $D_p < 200$ nm regardless of the biological activity within the seawater used. This was even though the SSA with $D_p < 200$ nm particles were likely highly enriched in organic matter during these mesocosms (Deane et al., Submitted), *and thus much smaller $\kappa_{app}$ would be expected*.

As to the note that we should include some references to support the statement "The often large CCN activation efficiency for particles observed in the marine environment, in particular nascent SSA particles," we mention here that we start the following sentence "For example…" and then two sentences later "As another example…". These are example references. However, to make clear that we do not only mean ambient measurements, we have modified the statement to now read: "A lowering of sc due to reductions in σ has been used (or hypothesized) to explain the often large CCN activation efficiency for particles observed in the marine environment *or in microcosm experiments*, in particular nascent SSA particles." We also note here the dependent clause "…in particular nascent SSA particles."

We thank the reviewer for pointing us to the Rasmussen et al. (2017) paper, which we had missed since the focus was on particle volatility. We have added the following to the discussion: "*Rasmussen et al. (2017) report negligible difference in CCN activity between artificial seawater, real seawater, and seawater spiked (at 1%-5% by volume) with sea surface microlayer samples. However, as with Collins et al. (2016), no composition measurements were made.*"

We appreciate the reference to Yakobi-Hancock. In the first part of this discussion we focus on primary SSA for this study, and the sulfate-to-organic ratio mentioned by the reviewer will be driven by secondary chemistry. While the cited Ovadnevaite et al. (2011) is for ambient measurements, and therefore likely includes some secondary processing, the "dichotomy" that

they observe provides evidence that there is something going on with their marine aerosols (e.g. surface tension depression) that goes beyond variability in e.g. the sulfate-to-organic ratio. However, this study is relevant in the context of the second Ovadnevaite et al. (2017) study cited. We have added brief discussion at that point.

> *Yakobi-Hancock et al. (2014) measured $\kappa_{app}$ values for ambient 50 nm and 100 nm particles on the west coast of Vancouver Island, sampling often marine influenced air. They observed substantial variability in the $\kappa_{app}$ values, partly attributable to variations in the sulfate-to-organic ratio but, we speculate, with at least some of the remaining scatter potentially resulting from variability in s depression at activation, depending on the exact chemical nature of the organic material.*

Looking at some of the references cited in Yakobi-Hancock et al. (2014), a limitation is that many of the potentially relevant studies investigated CCN activity of marine-impacted size distributions. For ambient measurements, it is important to focus on smaller particles, as for primary particles at least the organic content decreases with increasing size. Thus, we have not added further references.

Minor Title: The title says "Sea spray aerosols" – I suggest changing to "model sea spray", "salts" or "Sea spray aerosol mimics" or similar since the study was performed on NaCl core particles.

"Sea spray aerosols" was changed to "sea spray aerosol mimics" in the revised manuscript.

In the caption for Table 1 caption it says: "Uncertainties on the film model parameters on the precision in RH (see text for details)" – it is not clear to me what is meant here.

As stated on page 12 lines 2-4: The data were then fit to the film model using the perturbed (+ and -) *RH* values, and the uncertainties were calculated as the difference between the original and perturbed cases. To clarify this within the table, we have moved this statement from the caption to an expanded table footnote that now states "*Uncertainties on the film model fit parameters were estimated by performing separate fits after perturbing the input RH based on the precision-based uncertainty in the RH (see text for details).*"

Figure S2: it should say in the figure caption that the calibration was performed with NaCl particles.

This has been added to the revised manuscript.

Figure 2: figure text- "confidence interval" – is this 95% ? should be stated.

The confidence interval is indeed 95%. This is clarified in the updated manuscript.

**References:**

Bertram, A. K., Martin, S. T., Hanna, S. J., Smith, M. L., Bodsworth, A., Chen, Q., Kuwata, M., Liu, A., You, Y., and Zorn, S. R.: Predicting the relative humidities of liquid-liquid phase separation, efflorescence, and deliquescence of mixed particles of ammonium sulfate, organic material, and water using the organic-to-sulfate mass ratio of the particle and the oxygen-to-carbon elemental ratio of the organic component, Atmos. Chem. Phys., 11, 10995-11006, 10.5194/acp-11-10995-2011, 2011.

Collins, D. B., Bertram, T. H., Sultana, C. M., Lee, C., Axson, J. L., and Prather, K. A.: Phytoplankton blooms weakly influence the cloud forming ability of sea spray aerosol, Geophysical Research Letters, 43, 9975-9983, 10.1002/2016gl069922, 2016.

Deane, G. B., Cochran, R. E., Jayarathne, T., Rusch, L., Sultana, C. M., Lee, C., Tivanski, A. V., Cappa, C. D., Bertram, T. H., Prather, K. A., Grassian, V. H., and Stone, E. A.: Size-dependent transfer of organic matter to nascent sea spray aerosol, Submitted.

Mikhailov, E., Vlasenko, S., Niessner, R., and Pöschl, U.: Interaction of aerosol particles composed of protein and saltswith water vapor: hygroscopic growth and microstructural rearrangement, Atmospheric Chemistry and Physics, 4, 323-350, 10.5194/acp-4-323-2004, 2004.

Ovadnevaite, J., Ceburnis, D., Martucci, G., Bialek, J., Monahan, C., Rinaldi, M., Facchini, M. C., Berresheim, H., Worsnop, D. R., and O'Dowd, C.: Primary marine organic aerosol: A dichotomy of low hygroscopicity and high CCN activity, Geophysical Research Letters, 38, L21806, 10.1029/2011GL048869, 2011.

Ovadnevaite, J., Zuend, A., Laaksonen, A., Sanchez, K. J., Roberts, G., Ceburnis, D., Decesari, S., Rinaldi, M., Hodas, N., Facchini, M. C., Seinfeld, J. H., and O' Dowd, C.: Surface tension prevails over solute effect in organic-influenced cloud droplet activation, Nature, 546, 637-641, 10.1038/nature22806, 2017.

Petters, M., and Kreidenweis, S.: A single parameter representation of hygroscopic growth and cloud condensation nucleus activity, Atmospheric Chemistry and Physics, 7, 1961-1971, 10.5194/acp-7-1961-2007, 2007.

Petters, S. S., and Petters, M. D.: Surfactant effect on cloud condensation nuclei for two-component internally mixed aerosols, Journal of Geophysical Research: Atmospheres, 121, 1878-1895, doi:10.1002/2015JD024090, 2016.

Rasmussen, B. B., Nguyen, Q. T., Kristensen, K., Nielsen, L. S., and Bilde, M.: What controls volatility of sea spray aerosol? Results from laboratory studies using artificial and real seawater samples, Journal of Aerosol Science, 107, 134-141, 10.1016/j.jaerosci.2017.02.002, 2017.

Ruehl, C. R., Davies, J. F., and Wilson, K. R.: An interfacial mechanism for cloud droplet formation on organic aerosols, Science, 351, 1447-1450, 10.1126/science.aad4889, 2016.

Song, M., Marcolli, C., Krieger, U. K., Zuend, A., and Peter, T.: Liquid-liquid phase separation in aerosol particles: Dependence on O:C, organic functionalities, and compositional complexity, Geophysical Research Letters, 39, 10.1029/2012GL052807, 2012.

Topping, D., Barley, M., Bane, M. K., Higham, N., Aumont, B., Dingle, N., and McFiggans, G.: UManSysProp v1.0: an online and open-source facility for molecular property prediction and

atmospheric aerosol calculations, Geoscientific Model Development, 9, 899-914, 10.5194/gmd-9-899-2016, 2016.

Yakobi-Hancock, J. D., Ladino, L. A., Bertram, A. K., Huffman, J. A., Jones, K., Leaitch, W. R., Mason, R. H., Schiller, C. L., Toom-Sauntry, D., Wong, J. P. S., and Abbatt, J. P. D.: CCN activity of size-selected aerosol at a Pacific coastal location, Atmospheric Chemistry and Physics, 14, 12307-12317, 10.5194/acp-14-12307-2014, 2014.

Zelenyuk, A., Cai, Y., and Imre, D.: From Agglomerates of Spheres to Irregularly Shaped Particles: Determination of Dynamic Shape Factors from Measurements of Mobility and Vacuum Aerodynamic Diameters, Aerosol Science and Technology, 40, 197-217, 10.1080/02786820500529406, 2006.